# Agrin-Lrp4-Ror2 signaling regulates adult hippocampal neurogenesis in mice

Hongsheng Zhang[1], Anupama Sathyamurthy[2], Fang Liu[2], Lei Li[1], Lei Zhang[1], Zhaoqi Dong[1], Wanpeng Cui[1], Xiangdong Sun[2], Kai Zhao[2], Hongsheng Wang[1], Hsin-Yi Henry Ho[3], Wen-Cheng Xiong[1,2,4], Lin Mei[1,2,4]*

[1]Department of Neurosciences, School of Medicine, Case Western Reserve University, Cleveland, United States; [2]Department of Neuroscience and Regenerative Medicine, Medical College of Georgia, Augusta University, Augusta, United States; [3]Department of Neurobiology, Harvard Medical School, Boston, United States; [4]Louis Stokes Cleveland Veterans Affairs Medical Center, Cleveland, United States

**Abstract** Adult neurogenesis in the hippocampus may represent a form of plasticity in brain functions including mood, learning and memory. However, mechanisms underlying neural stem/progenitor cells (NSPCs) proliferation are not well understood. We found that Agrin, a factor critical for neuromuscular junction formation, is elevated in the hippocampus of mice that are stimulated by enriched environment (EE). Genetic deletion of the *Agrn* gene in excitatory neurons decreases NSPCs proliferation and increases depressive-like behavior. Low-density lipoprotein receptor-related protein 4 (Lrp4), a receptor for Agrin, is expressed in hippocampal NSPCs and its mutation blocked basal as well as EE-induced NSPCs proliferation and maturation of newborn neurons. Finally, we show that Lrp4 interacts with and activates receptor tyrosine kinase-like orphan receptor 2 (Ror2); and *Ror*2 mutation impairs NSPCs proliferation. Together, these observations identify a role of Agrin-Lrp4-Ror2 signaling for adult neurogenesis, uncovering previously unexpected functions of Agrin and Lrp4 in the brain.

DOI: https://doi.org/10.7554/eLife.45303.001

*For correspondence:
lin.mei@case.edu

Competing interests: The authors declare that no competing interests exist.

## Introduction

Brains change their structure and function in response to environmental alterations. One such adaptation mechanism is to form new neurons and circuits. In rodents, the hippocampus forms more newborn neurons when animals are subjected to housing in an enriched environment, voluntary running exercise, special task learning, electroconvulsive stimulation or antidepressant treatment (*Bond et al., 2015*; *Gonçalves et al., 2016*; *Kempermann, 2015*). Adult neurogenesis is implicated in learning, memory and mood regulation and its decline is thought to contribute mood and cognitive deficits of aging and Alzheimer's disease (*Ming and Song, 2011*; *Mu and Gage, 2011*). Adult neurogenesis has been observed in various species including human, monkeys, and rodents (*Altman and Das, 1965*; *Eriksson et al., 1998*; *Gould et al., 1999*), although a recent paper reported that neurogenesis in healthy human brains might be more conserved than previously thought (*Boldrini et al., 2018*; *Kempermann et al., 2018*; *Sorrells et al., 2018*).

In adult hippocampus, NSPCs line in the sub-granule zone (SGZ) and proliferate to generate newborn neurons that integrate into the granule cell layer of the dentate gyrus (DG) (*Gonçalves et al., 2016*). This dynamic process includes quiescent stem cell activation, proliferation, neuronal fate specification, migration and synaptic integration (*Ming and Song, 2011*). The balance between neural stem cells (NSCs) quiescence and proliferation is regulated tightly because a paucity of proliferating NSCs would produce too few new neurons, and excessive proliferation could deplete the neural

progenitor cells (NPCs) pool (*Cheung and Rando, 2013*). Recent studies have shed light on molecular mechanisms controlling the balance. For example, BMP and Notch have been shown to be necessary for maintaining NSCs quiescence (*Ables et al., 2010*; *Ehm et al., 2010*; *Mira et al., 2010*), while NSCs proliferation could be promoted by Wnt, IGF1, and VEGF (*Bracko et al., 2012*; *Han et al., 2015*; *Jang et al., 2013*; *Qu et al., 2010*; *Seib et al., 2013*). Nevertheless, molecular mechanisms controlling this balance remain unclear.

Agrin is a proteoglycan utilized by motoneurons for postsynaptic assembly of the neuromuscular junction (*McMahan, 1990*). It acts by binding to Lrp4, a single transmembrane protein of the low-density-lipoprotein (LDL) family, and thus activates the receptor tyrosine kinase MuSK (*Kim et al., 2008*; *Zhang et al., 2008*). Ensuing signaling leads to multiple events including concentration of acetylcholine receptors and presynaptic differentiation and eventual formation of the peripheral synapse (*Li et al., 2018*). Both Agrin and Lrp4 are expressed in the brain (*Gesemann et al., 1998*; *Sun et al., 2016*). Interestingly, Lrp4 is expressed in adult hippocampal NSPCs, and the level is decreased with progression of newborn neurons (*Habib et al., 2016*; *Shin et al., 2015*). We posit that Agrin and Lrp4 regulate adult hippocampal neurogenesis. To test this hypothesis, we first determined whether EE alters the expression of Agrin and Lrp4 and investigated if adult neurogenesis requires Agrin and Lrp4 by neuron- and astrocyte-specific mutation and in adult NSPCs. Second, we determined whether *Lrp4* mutation alters maturation of newborn neurons and which domains of Lrp4 are necessary. Third, we investigated how Lrp4 regulates adult neurogenesis. Our results suggest a working model where Agrin via Lrp4 activates the receptor tyrosine kinase Ror2 to promote adult neurogenesis.

## Results

### Requirement of EE-induced Agrin in adult neurogenesis

To identify factors that contribute to EE-induced neurogenesis in the hippocampus, we adopted an EE behavioral paradigm as previously described (*Sztainberg and Chen, 2010*) (*Figure 1A*). Mice were housed for 4 weeks in a chamber ($86 \times 76 \times 24$ cm) that contained two running wheels, tubes and nest boxes (designated as EE cage). Compared with mice that were housed in standard cages (SC), mice housed in EE cages displayed more Arc$^+$ granule cells in the dental gyrus region of the hippocampus (*Figure 1B and C*), in agreement with previous reports (*Pinaud et al., 2001*). To validate this behavioral paradigm, we analyzed hippocampal mRNA for expression of various secretable proteins. EE increased levels of *Bdnf*, *Igf*1, and *Vegf* (*Figure 1D*), in agreement with previous reports (*Cao et al., 2004*; *Keyvani et al., 2004*; *Rossi et al., 2006*). Unexpectedly, *Agrn* was also increased in the hippocampus of EE animals, as compared with SC animals. This effect appeared to be specific because levels of *ApoE* and *Wnt5a* remained similar between mice of EE cages and SC (*Figure 1D*). Interestingly, the expression level of *Lrp4*, a receptor of Agrin, was increased by EE. In contrast, expression of *MuSK*, which was low in the brain, was not changed by EE (*Figure 1E*). These results led us to posit that Agrin, possibly via Lrp4, may contribute to EE-induced adult neurogenesis in the hippocampus.

To test this hypothesis, we generated neuron-specific *Agrn* knockout mice by crossing *Agrn*$^{f/f}$ mice with *Neurod*6-*Cre* mice where *Cre* is expressed under the promoter of the gene of *Neurod*6 (*Goebbels et al., 2006*). Neurod6 is a transcription factor whose expression in mice is specific in neurons and begins at E11.5 (*Goebbels et al., 2006*). Resulting *Neurod*6-*Cre;Agrn*$^{f/f}$ (referred as Agrin CKO) had ~50% reduction in total *Agrn* mRNA levels in the hippocampus, compared with control mice (*Figure 1—figure supplement 1A and B*). *Agrn* has two isoforms: neuronal *Agrn* and non-neuronal *Agrn* (*Li et al., 2018*). The residual *Agrn* mRNA in Agrin CKO mice may come from non-neuronal cells. Indeed, by using primers specific for neuronal *Agrn*, 80% reduction was observed in Agrin CKO hippocampus (*Figure 1—figure supplement 1B*), indicating a specific ablation of neuronal *Agrn* (hereafter referred as *Agrn*). Agrin CKO mice had similar body weight to control mice (*Agrn*$^{f/f}$ or *Agrn*$^{f/+}$ mice) (*Figure 1—figure supplement 1C*) and did not exhibit global morphological deficits. In particular, hippocampal structures of Agrin CKO mice were similar to those of control mice (*Figure 1—figure supplement 1D and E*).

To determine whether Agrin is indispensable for adult neurogenesis, we injected BrdU into mice to label proliferating cells in the hippocampus as previously described (*Appel et al., 2018*). The

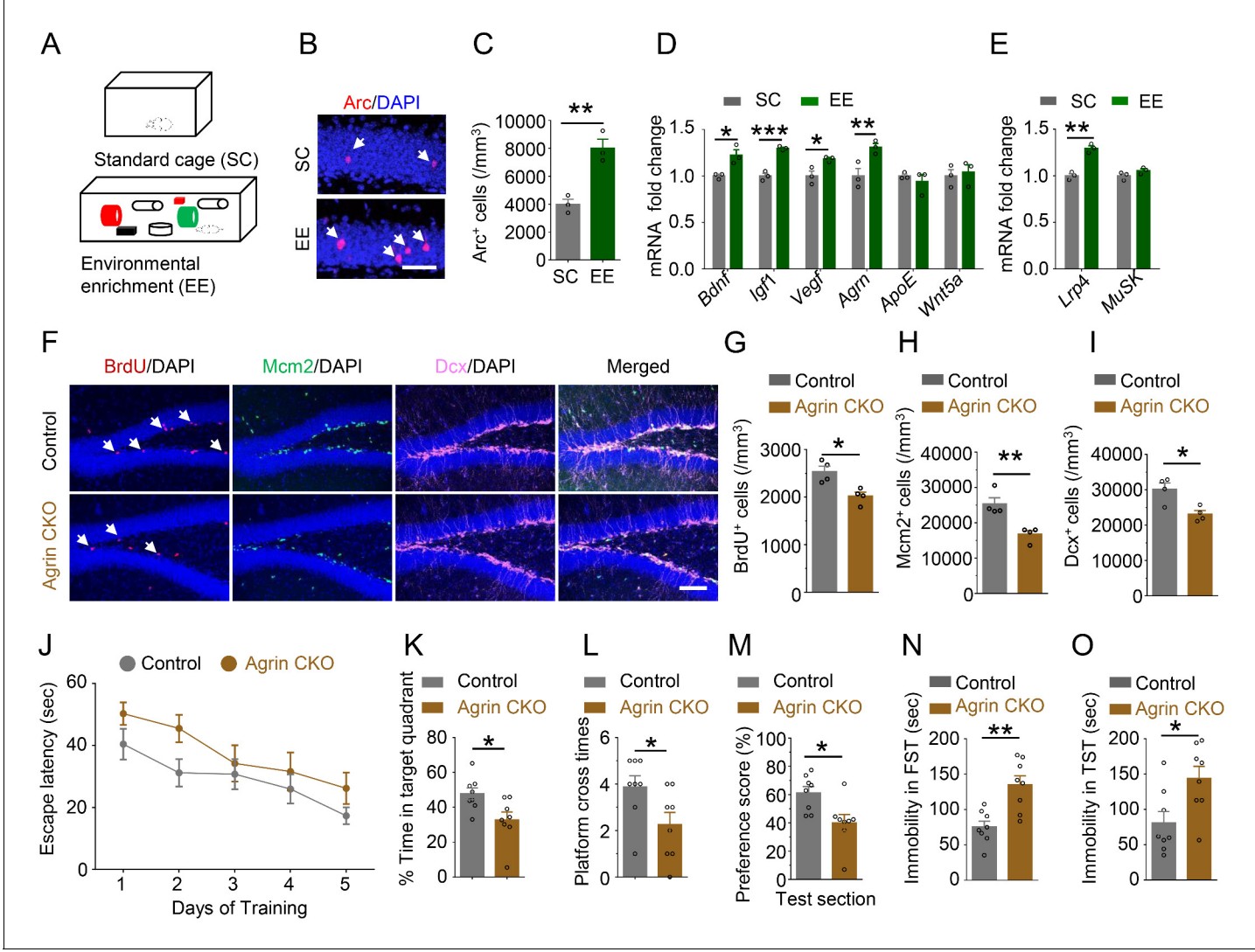

**Figure 1.** Requirement of Agrin for adult hippocampal neurogenesis. (**A**) Schematic diagram of standard cage (SC) and environmental enrichment (EE) housing. (**B–C**) Increased Arc+ cells in hippocampus of mice in EE, compared with SC-housed mice. n = 3 for each group, Student's t-test: t (4)=5.493, p=0.0054. (**D**) Increased *Agrn* mRNA level in hippocampus of EE-housed mice, compared with SC-housed mice. n = 3 for each group. Student's t-test: t (4)=3.641, p=0.022 (*Bdnf*); t (4)=9.545, p=0.0007 (*Igf*1); t (4)=3.32, p=0.0294 (*Vegf*); t (4)=3.434, p=0.0264 (*Agrn*); t (4)=0.7758, p=0.4812 (*ApoE*); t (4) =0.3968, p=0.7117 (*Wnt*5a). (**E**) Increased *Lrp4* mRNA level in hippocampus of EE-housed mice, compared with SC-housed mice. n = 3 for each group. Student's t-test: t (4)=8.04, p=0.0013 (*Lrp4*); t (4)=1.76, p=0.1527(*MuSK*). (**F–I**) Reduced BrdU, Mcm2, and Dcx-labeled cells in Agrin CKO hippocampal SGZ. (**F**) Representative images. Scale bar,100 µm. (**G–I**) Stereological quantification of BrdU+ (**G**), Mcm2+ (**H**), and Dcx+ (**I**) cells. n = 4 for each group. Student's t-test: t (6)=3.656, p=0.0106 for BrdU; t (6)=4.185, p=0.0058 for Mcm2; t (6)=3.410, p=0.0143 for Dcx. (**J**) Agrin CKO mice increased latency to find the hidden platform F(1,70)=7.81, p=0.0067. (**K**) Reduced time spent in target quadrant. n = 8 for each group, Student's t test: t (14)=2.639, p=0.0195. (**L**) Reduced number of platform crossings. n = 8 for each group, Student's t test: t (14)=0.0386. (**M**) Reduced preference score during test section. n = 8 for each group, Student's t test t (14)=2.865, p=0.0125. (**N–O**) Increased immobility of Agrin CKO mice, compared with control mice, in FST (**N**) and TST (**O**). n = 8 for each group. Student's t-test: t (14)=3.956, p=0.0014 for FST; t (14)=2.691, p=0.0175 for TST. Data are mean ± s.e.m; *, p<0.05; **, p<0.01; ***, p<0.001.

DOI: https://doi.org/10.7554/eLife.45303.002

The following source data and figure supplement are available for figure 1:

**Source data 1.** Requirement of Agrin for adult hippocampal neurogenesis.
DOI: https://doi.org/10.7554/eLife.45303.004
**Source data 2.** Characterization of neuronal *Agrn* knockout mice.
DOI: https://doi.org/10.7554/eLife.45303.005
**Figure supplement 1.** Generation and characterization of neuronal *Agrn* knockout mice.
DOI: https://doi.org/10.7554/eLife.45303.003

density of BrdU⁺ cells was decreased in Agrin CKO mice compared with littermate controls, suggesting a compromised cell proliferation in mutant hippocampus. To further test the hypothesis, we stained the hippocampus for Mcm2, a marker of cell proliferation, and Dcx, a marker of immature neurons; and found the density of both Mcm2⁺ and Dcx⁺ cells were decreased (*Figure 1F–1I*). These results suggest that Agrin may be indispensable for adult hippocampal neurogenesis. Impaired adult hippocampal neurogenesis has been shown to correlate with memory and mood in mice (*Anacker and Hen, 2017*). Therefore, Agrin CKO mice were subjected to a battery of behavioral test. In the training phase of Morris water maze, the escape latency for *Agrn* mutant mice to locate the hidden platform was increased, compared with that of control mice (*Figure 1J*). The mutant mice exhibited similar swimming speed as control mice (*Figure 1—figure supplement 1F*). During the probe test, *Agrn* mutant mice spent less time in the platform quadrant and exhibited fewer crosses over the absent platform (*Figure 1K and L*). These results suggest that *Agrn* mutant mice may be impaired in learning and memory. This notion was further supported by lower preference scores in object location test (*Figure 1M*, *Figure 1—figure supplement 1G*). In the forced swimming test (FST) and tail suspension test (TST), Agrin CKO mice increased the duration of immobility (*Figure 1N and O*), suggesting depressive-like behavior in *Agrn* mutant mice. Together, these observations suggest that excitatory neurons in DGs expresses Agrin, which promotes adult hippocampal neurogenesis.

## Lrp4 for adult hippocampal NSPCs proliferation

*Lrp*4 mRNA was increased in hippocampus by EE (*Figure 1E*). It would be important to determine in which cells Lrp4 is expressed. Unfortunately, currently available anti-Lrp4 antibodies were not good for immunostaining. To this end, we characterized β-gal expression in the hippocampus of *Lrp4-LacZ* reporter mice (*Sun et al., 2016*). In this strain, the *Lrp*4 gene (exons 2–30) was replaced by a cassette containing the *LacZ* gene. Under the control of the endogenous promoter of *Lrp*4, β-gal activity is believed to faithfully indicate the expression pattern of *Lrp*4. As shown in *Figure 2A*, β-gal was enriched in cells in SLM and ML layers of the hippocampus, which were mostly astrocytes (*Sun et al., 2016*). Interestingly, β-gal was also detected in the SGZ of the DG (*Figure 2A*), which NSPCs reside. To determine in what cells β-gal is expressed in the SGZ, sections were co-stained with antibodies against β-gal and markers of NSCs and derivatives at different stages (*Ming and Song, 2011*). As shown in *Figure 2B*, β-gal activity was detected in cells labeled by Nestin, a marker of neural stem cells and progenitor cells (*Ming and Song, 2011*). In addition, β-gal⁺ cells were positive for Gfap, a marker of radial glia-like cells (RGLs) (*Ming and Song, 2011*). However, β-gal activity was barely detectable in cells positive for Tbr2 (*Figure 2C*), a marker of progenitor cells (*Ming and Song, 2011*) or PSA-NCAM (*Figure 2D*), a marker of immature neurons. These results indicate that Lrp4 is expressed in precursor cells including RGLs, intermediate progenitor cells, and neuroblasts. These results are consistent with recent single-cell RNA-seq results that *Lrp*4 is expressed in astrocytes, RGLs and progenitors, but not in more mature neurobloasts or dentate granule neurons (*Habib et al., 2016*; *Hochgerner et al., 2018*; *Shin et al., 2015*).

To determine whether Lrp4 plays a role in adult hippocampal neurogenesis, *Lrp*4 knockout mice were generated by crossing *Lrp4^{f/f}* mice with h*GFAP-Cre* mice (*Figure 3—figure supplement 1A*) where *Cre* is under the promoter of the human *GFAP* gene (*Zhuo et al., 2001*). Lrp4 expression was abolished in the brain in resulting *GFAP-Cre::Lrp4^{f/f}* (Lrp4 CKO) (*Figure 3—figure supplement 1B–1D*). Remarkably, BrdU⁺ cells were reduced in Lrp4 CKO hippocampus (*Figure 3A and B*), suggesting an indispensable role of Lrp4 in maintaining adult neurogenesis. Consequently, without Lrp4, Dcx⁺ cells were fewer in Lrp4 CKO dentate gyrus (*Figure 3A and C*). BrdU⁺ cells reduction may result from a diminished pool of quiescent neural stem cells and/or reduced numbers of proliferating stem cells including activated neural stem cells, progenitor cells, and neuroblast cells (*Kempermann et al., 2015*). To test this, we characterize the number of cells that are positive for Gfap and Sox2, a marker of neural stem cells and progenitor cells. Cells positive for these two markers are quiescent neural stem cells (*Bonaguidi et al., 2011*; *Encinas et al., 2011*). As shown in *Figure 3D and E*, the number of Gfap⁺Sox2⁺BrdU⁺ cells was similar between control and Lrp4 CKO mice, suggesting that *Lrp*4 knockout may not affect neural stem cell division. However, *Lrp*4 mutation reduced the number of BrdU⁺Sox2⁺ cells (*Figure 3F*), indicating that Lrp4 is indispensable for progenitor cell proliferation in the SGZ. The reduction of BrdU⁺ Sox2⁺ cell was not due to increased cell death because there was no difference between apoptotic cells positive for cleaved caspase-3

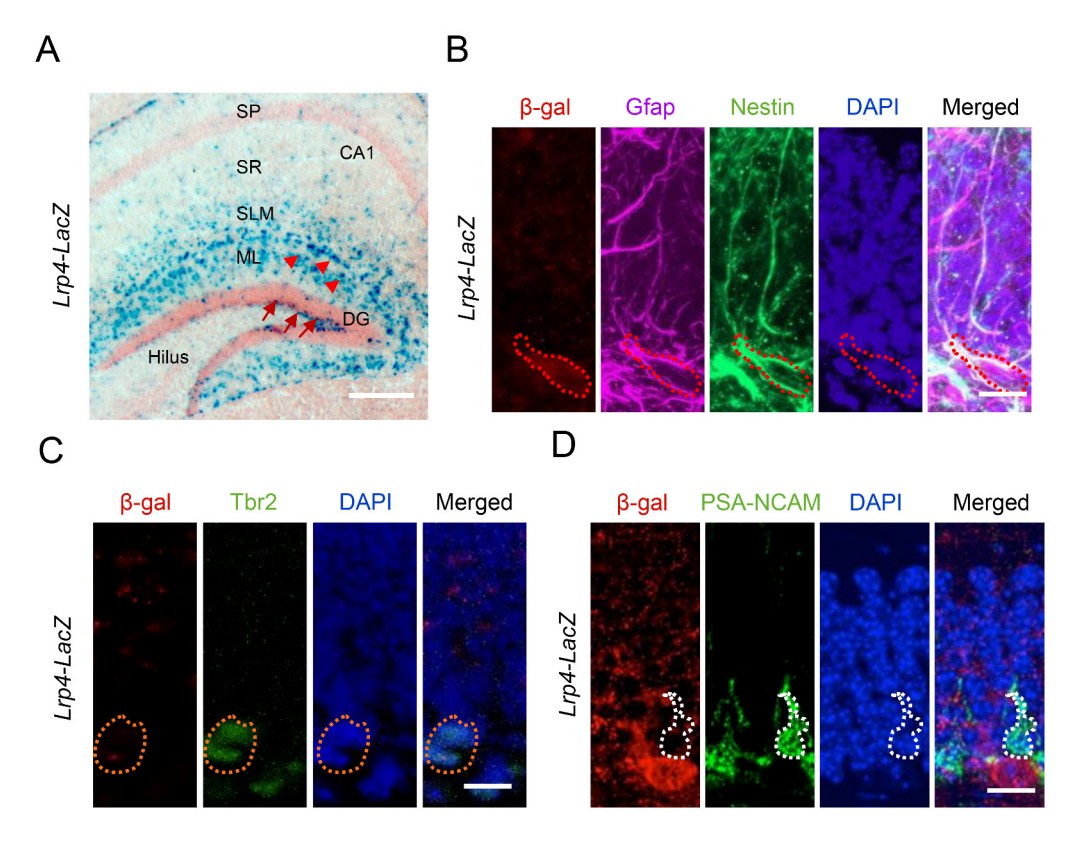

**Figure 2.** Lrp4 expression in adult hippocampal NSPCs in mice. (A) X-gal staining of coronal brain sections of *Lrp4*-LacZ mice. Arrowheads, astrocytes; arrows, NSPCs. Scale bar, 100 µm. (B) Lrp4 expression in neural stem cells labeled by Gfap and Nestin. DG sections of *Lrp4-LacZ* mice were stained for β-gal, Gfap, Nestin, and DAPI. A representative cell was circled that was positive for Gfap and Nestin. Scale bar, 5 µm. (C, D) No-detectable β-gal level in Tbr2[+] (C) and PSA-NCAM (D) cells in DG. Scale bar, 5 µm.

DOI: https://doi.org/10.7554/eLife.45303.006

between control and Lrp4 CKO mice (*Figure 3—figure supplement 2A and B*). In agreement with decreased adult hippocampal neurogenesis, Lrp4 CKO mice showed increased immobility in FST and TST (*Figure 3G and H*). Together, these results identify a critical role of Lrp4 in adult hippocampal neurogenesis.

## Lrp4 cell-autonomous regulation of EE-induced adult neurogenesis

Early-stage Gfap[+] cells can develop into neurons as well as astrocytes (*Noctor et al., 2001*). Because Lrp4 is expressed in astrocytes in developed brains (*Sun et al., 2016*), we determined whether Lrp4 in NSPCs is indispensable by crossing *Lrp4[f/f]* mice with *Nes-Cre/ERT2;Ai9* mice. In *Nes-Cre/ERT2* mice, Cre is expressed in NSPCs, but its activity is inactive until induction by tamoxifen (Tam) (*Lagace et al., 2007*). *Ai9* mice carry floxed tdTomato cassette in the Gt(ROSA)26Sor locus and express tdTomato in a *Cre*-dependent manner upon Tam induction. Resulting *Nes-Cre/ERT2::Ai9:: Lrp4[f/f]* mice were injected with Tam (referred as Nes Lrp4 CKO mice). *Lrp*4 mRNA and protein were reduced in the dentate gyrus of Nes Lrp4 CKO mice, compared with Tam-treated *Nes-Cre/ERT2:: Ai9::Lrp4[+/+]* mice (referred as control) (*Figure 4—figure supplement 1A–1E*). tdTomato-labeled cells were reduced in dentate gyrus 2 days as well 1 month after Tam injection (*Figure 4A–4E*). The reduced number of tdTomato-labeled cells not due to increased cell death because there was no difference of apoptotic cells positive for cleaved caspase-3 between control mice and Nes Lrp4 CKO mice 2 days after Tam treatment (*Figure 4—figure supplement 1F and G*). Because tdTomato expression was controlled by *Nes-Cre*, these results provide further support to the hypothesis that

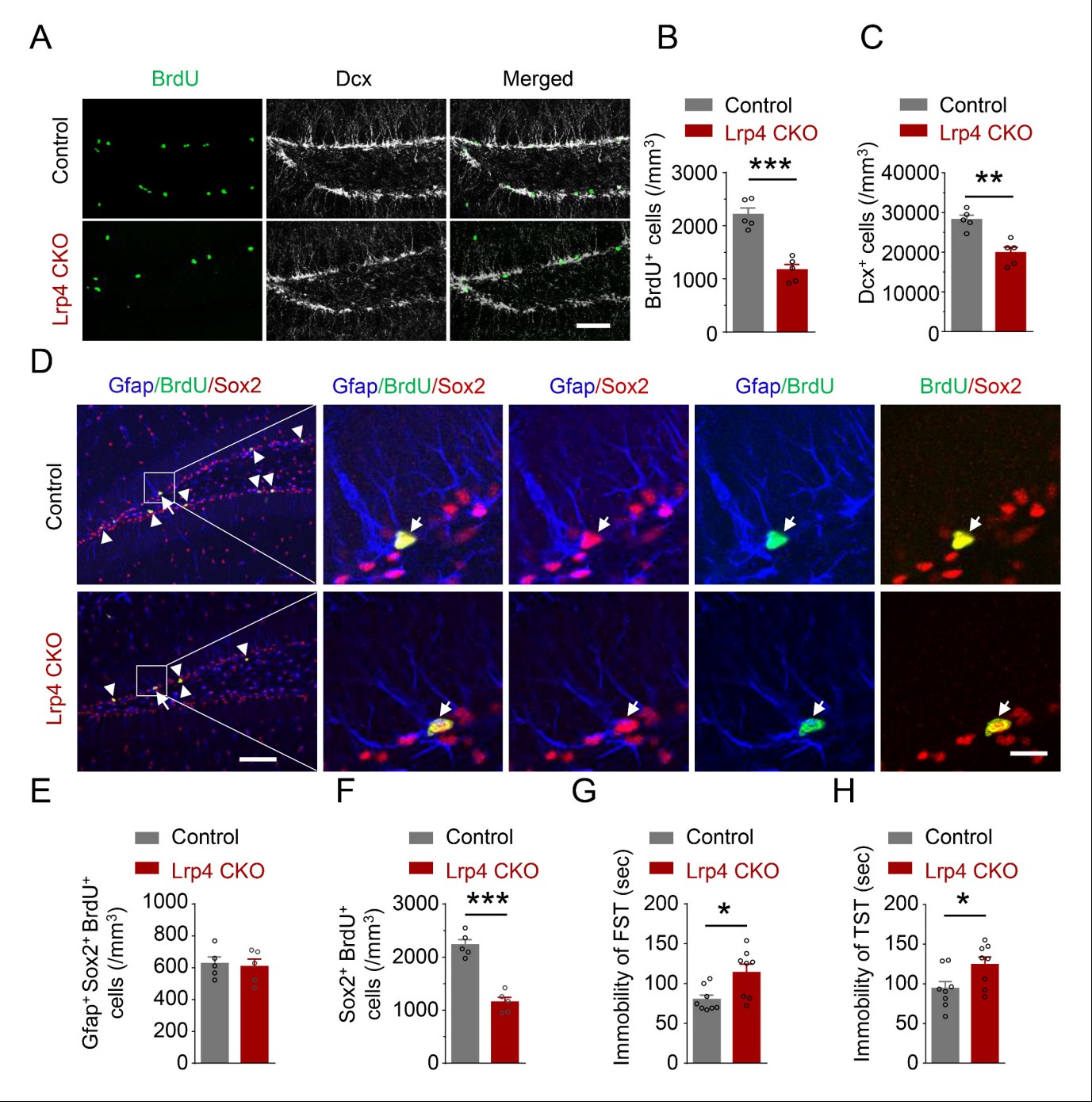

**Figure 3.** Reduced NSPCs proliferation and increased immobility of Lrp4 CKO mice. (A–C) Reduced numbers of BrdU- and Dcx-labeled cells in Lrp4 CKO SGZ. (A) Representative images. Scale bar, 100 μm. (B–C) Stereological quantification of SGZ BrdU$^+$ (B) and Dcx$^+$ (C) cells. n = 5 for each group. Student's t-test: t (8)=6.602, p=0.0002 for BrdU; t (8)=4.701, p=0.0015 for Dcx. (D–F) Reduced NPCs proliferation in Lrp4 CKO. (D) Representative images. The arrow indicated Gfap$^+$/BrdU$^+$/Sox2$^+$, while the arrow head indicated BrdU$^+$/Sox2$^+$ cells. Scale bar, left 100 μm, right 20 μm. (E) Similar numbers of SGZ Gfap$^+$Sox2$^+$ BrdU$^+$ NSCs between the two genotypes. n = 5 for each group. Student's t-test: t (8)=0.2947, p=0.7757. (F) Decreased the density of Sox2$^+$BrdU$^+$ NPCs in Lrp4 CKO mice, compared with control. n = 5 for each group. Student's t-test: t (8)=7.943, p<0.0001. (G–H) Increased duration of immobility in FST (G) and TST (H) of Lrp4 CKO mice, compared with control. n = 8 for each group. Student's t-test: t (14)=2.826, p=0.0135 for FST; t (14)=2.332, p=0.0352 for TST. Data are mean ± s.e.m; *, p<0.05; **, p<0.01; ***, p<0.001.

DOI: https://doi.org/10.7554/eLife.45303.007

The following source data and figure supplements are available for figure 3:

**Source data 1.** Reduced adult neurogenesis and increased immobility of Lrp4 CKO mice.

*Figure 3 continued on next page*

*Figure 3 continued*

DOI: https://doi.org/10.7554/eLife.45303.010

**Source data 2.** Characterization of Lrp4 CKO mice.

DOI: https://doi.org/10.7554/eLife.45303.011

**Source data 3.** Similar number of cleaved caspase-3 labeled cells between Lrp4 CKO and control mice.

DOI: https://doi.org/10.7554/eLife.45303.012

**Figure supplement 1.** Generation and characterization of *Lrp*4 mutant mice.

DOI: https://doi.org/10.7554/eLife.45303.008

**Figure supplement 2.** Similar number of cleaved caspase-3 labeled cells between Lrp4 CKO and control mice.

DOI: https://doi.org/10.7554/eLife.45303.009

Lrp4 in NSPCs is critical and Lrp4 regulates adult neurogenesis in a cell-autonomous manner and at basal level more likely at the progenitor level.

To determine whether NSPCs Lrp4 contributes to EE-induced adult neurogenesis, we housed Nes Lrp4 CKO and control mice in EE cages for 4 weeks after Tam treatment (*Figure 4F*). In control mice, EE increased the numbers of Ki67$^+$ cells (*Figure 4G and H*), suggesting an increase in cell pro- liferation in the DG. Similar increase was observed with BrdU$^+$ cells (*Figure 4I and J*). To determine whether the increase occurred in NSCs and/or NPCs, we quantified BrdU$^+$ cells in Gfap$^+$Sox2$^+$ and Sox2$^+$ populations, respectively. Both were increased by EE (*Figure 4K and L*), in agreement with previous results (*Meshi et al., 2006*). Interestingly, the density of BrdU$^+$Gfap$^+$Sox2$^+$ cells were simi- lar in Nes Lrp4 CKO mice, compared with control, suggesting that *Lrp*4 knockout does not change the proliferation of NSCs at SC. However, the density of BrdU$^+$Sox2$^+$ cells was reduced at SC by *Lrp*4 knockout, suggesting that Lrp4 is necessary for NPCs proliferation (*Figure 4L*). In either case, EE-induced increase in BrdU$^+$ cells was attenuated by *Lrp*4 mutation (*Figures 4J, K and L*). These results suggest that Lrp4 in NSPCs is involved in EE-induced adult neurogenesis. In accord, Nes Lrp4 CKO mice displayed increased immobility in FST and TST when housed in SC cages, compared with control mice (*Figure 4M and N*). In addition, unlike control mice that showed EE-induced decrease in immobility, Nes Lrp4 CKO mice failed to respond to EE (*Figure 4M and N*). Together, these observations indicate that ablation of *Lrp*4 from NSPCs blocked EE-induced adult neurogenesis and behavioral improvement and suggest that Lrp4 regulates NSPCs proliferation in a cell-autonomous manner.

In addition to NSPCs proliferation, EE has been implicated in integration of newborn neurons into adult dentate gyrus (*Chancey et al., 2013*). To test whether this process requires Lrp4, we examined dendritic growth of newborn neurons in adult mice. To label proliferating NSPCs, *Lrp*4$^{f/f}$ mice were injected with retroviruses expressing GFP-fused wild-type Cre (*Cre-GFP*) and inactive Cre (*D-Cre-GFP*) (*Figure 4—figure supplement 2A*). GFP-labeled progenies were subjected to morphology analysis. As shown in *Figure 4—figure supplement 2B*, dendrites of D-Cre-GFP$^+$ neurons were extensively arborized at 28 dpi. In contrast, Cre-GFP$^+$ neurons (where *Lrp*4 was ablated) showed less arborization. Branch number, total length, and complexity of dendrites were reduced in Cre-GFP$^+$ neurons, compared with D-Cre-GFP$^+$ neurons (*Figure 4—figure supplement 2C–2F*). Similar den- drite deficits were observed at 42 dpi when newborn neurons are mature and fully integrated into the circuity. We also examined spines at these time points and found that *Lrp*4 ablation reduced spine density (*Figure 4—figure supplement 2G–2I*), suggesting compromised dendritic spine for- mation. Together, these data indicate that Lrp4 is necessary for dendritic arborization and spine for- mation in newly generated neurons in adult dentate gyrus and this effect in cell autonomous manner.

## Lrp4 as a receptor for Agrin to activate Ror2

In NMJ formation and maintenance, Lrp4 serves as a receptor for Agrin to activate the transmem- brane kinase MuSK (*Li et al., 2018*). The requirement of Agrin and Lrp4 for EE-induced adult neuro- genesis suggests that they work together. The β1 propeller domain of Lrp4 is required for and sufficient to mediate interaction with Agrin. On the other hand, the β3 propeller domain was shown to interact with MuSK and to be necessary for activating the kinase (*Zhang et al., 2011*). To deter- mine whether these domains are indispensable for EE-induced adult neurogenesis, we generated transgenic mice carrying loxP-STOP-loxP (LSL)-Flag-Lrp4Δβ1 or Flag-Lrp4Δβ3 and crossed them with

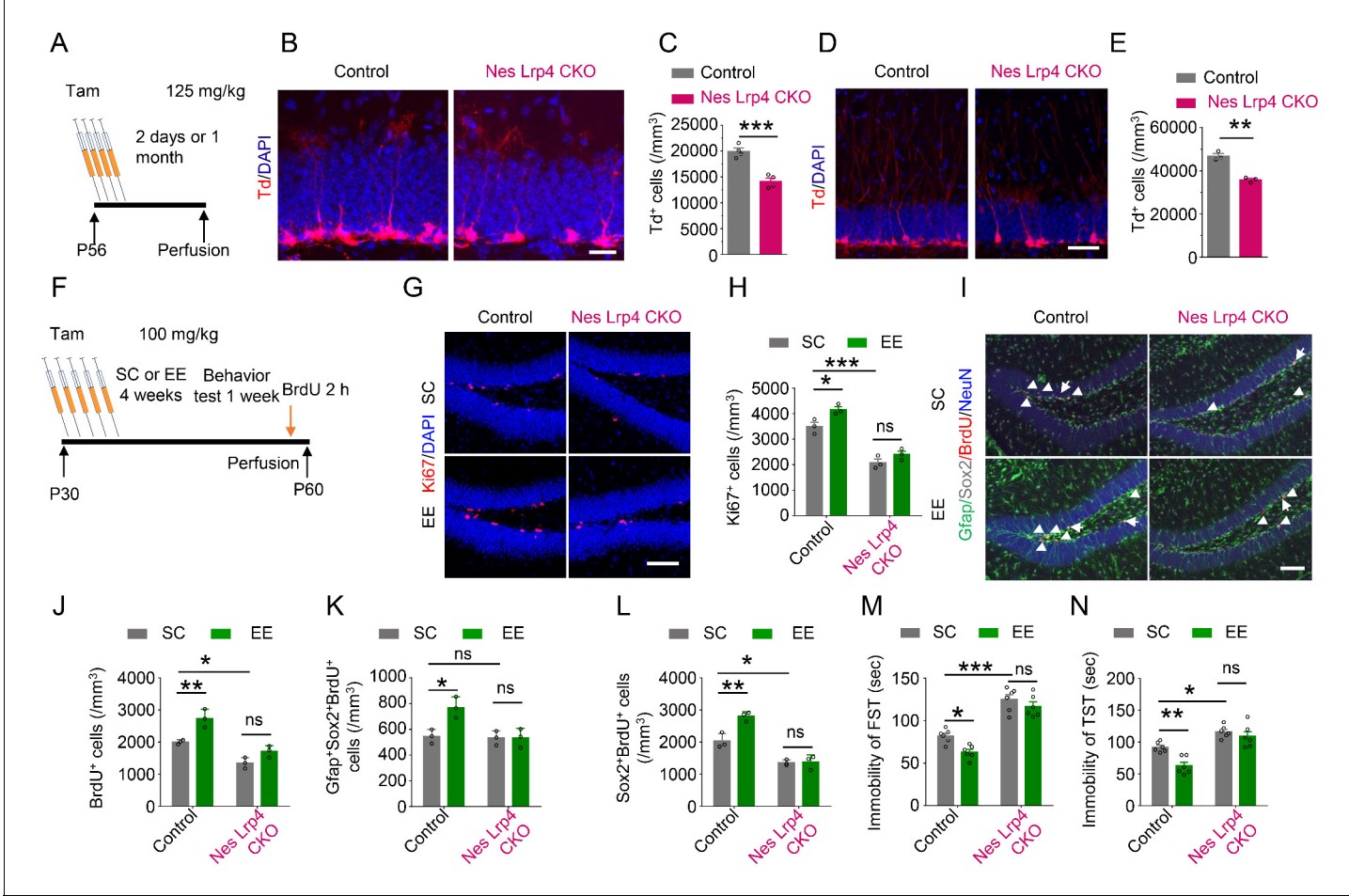

**Figure 4.** Cell-autonomous effect of Lrp4 in regulating NSPCs proliferation and behavior improvement. (**A**) The protocol of Tam treatment. (**B–C**) Decreased Td+ cells in Nes Lrp4 CKO mice compared with control at 2 days after Tam treatment. (**B**) Representative images. Scale bar, 25 μm. (**C**) Stereological quantification of Td+ cell density. n = 4 for each group. Student's t-test: t (6)=6.553, p=0.0006. (**D–E**) Decreased Td+ cells in Nes Lrp4 CKO mice compared with control after 1 months of Tam treatment. (**D**) Representative images. Scale bar, 50 μm. (**E**) Stereological quantification of Td+ cell density. Student's t-test: t (4)=8.159, p=0.0012. (**F**) Time schedule of Tam injection, EE, and BrdU administration. (**G–H**) EE for 4 weeks failed to increase the density of Ki67+ cells in the DG of iNestin-Lrp4$^{f/f}$ mice. (**G**) Representative images, Scale bar, 100 μm. (**H**) Stereological quantification of Ki67+ cell density. n = 3 for each group. Two-way ANOVA test, F (1,8)=129.4, p<0.0001 for genotype; F(1,8) = 12.4, p=0.0078 for EE). (**I–L**) EE for 4 weeks failed to increase the density of BrdU+, Gfap+Sox2+BrdU+, Sox2+BrdU+ cells in the DG of Nes Lrp4 CKO mice. (**I**) Representative images, Scale bar, 100 μm. (**J**) EE for 4 weeks failed to increase the density of BrdU+ cells in the DG of Nes Lrp4 CKO mice. n = 3 for each group. Two-way ANOVA test, F (1,8)=57.01, p<0.0001 for genotype; F (1,8)=24.28, p=0.0012 for EE. (**K**) EE for 4 weeks increase the density of Gfap+Sox2+BrdU+ cells in the DG of Nes Lrp4 CKO mice. n = 3 for each group. Two-way ANOVA test, F (1,8)=10.09, p=0.0131 for genotype; F(1,8)=8.321, p=0.0204 for EE. (**L**) EE for 4 weeks increase the density of Sox2+BrdU+ cells in the DG of Nes Lrp4 CKO mice. n = 3 for each group. Two-way ANOVA test, F (1,8)=98.21, p<0.0001 for genotype; F(1,8)=14.09, p=0.0056 for EE. (**M–N**) Nes Lrp4 CKOmice did not display decrease the duration of immobility in FST (**K**) and TST (**L**) after EE for 4 weeks. n = 6 for each group. In FST, two-way ANOVA test: F (1,20)=115.5, p<0.0001 for genotype; F (1,20)=9.04, p=0.007 for EE. In TST, two-way ANOVA test: F (1,20)=44.99, p<0.0001 for genotype; F (1,20)=11.85, p=0.0026 for EE. Data are mean ± s.e.m; ns, p>0.05; *, p<0.05; **, p<0.01; ***, p<0.001.

DOI: https://doi.org/10.7554/eLife.45303.013

The following source data and figure supplements are available for figure 4:

**Source data 1.** Cell-autonomous effect of Lrp4 in regulating NSPCs proliferation and behavior improvement.
DOI: https://doi.org/10.7554/eLife.45303.016
**Source data 2.** Characterization of inducible NSPCs-specific *Lrp*4 knockout mice.
DOI: https://doi.org/10.7554/eLife.45303.017
**Source data 3.** Impaired maturation of *Lrp*4 mutant newborn neurons.
DOI: https://doi.org/10.7554/eLife.45303.018
**Figure supplement 1.** Generation and characterization of inducible NSPCs-specific *Lrp*4 knockout mice.

*Figure 4 continued on next page*

*Figure 4 continued*

DOI: https://doi.org/10.7554/eLife.45303.014

**Figure supplement 2.** Impaired maturation of *Lrp4* mutant newborn neurons.

DOI: https://doi.org/10.7554/eLife.45303.015

Lrp4 CKO mice (*Figure 5A*, *Figure 5—figure supplement 1A–1D*). Transgenes under the control of the LSL cassette are not expressed until the STOP signal is floxed out (*Zinyk et al., 1998*). As shown in *Figure 5—figure supplement 1E*, Lrp4 revealed by anti-Lrp4 antibody was present in brains of control mice, but not of Lrp4 CKO mice. Flag-Lrp4Δβ1 and -Lrp4Δβ3 were revealed by anti-Flag antibody. They were not detectable in brains of *LSL-Lrp4β1* and *LSL-Lrp4β3* mice, but became detectable with anti-Flag antibody in brains of *GFAP-Cre::Lrp4^{f/f}::LSL- Lrp4β1* (Lrp4 CKO Δβ1) and *GFAP-Cre::Lrp4^{f/f}::LSL-Lrp4β3* (Lrp4 CKO Δβ3) mice. Notice that in these mice, Lrp4 was deleted in Lrp4 CKO brain (*Figure 5—figure supplement 1E*). As shown in *Figure 5B and D*, GFAP-mediated *Lrp4* deletion decreased the density of Dcx^+ and BrdU^+ cells (*Figure 5B–5E*). These deficits remained in mice expressing Lrp4Δβ1, indicating that the β1 domain is indispensable in Lrp4-regulated adult neurogenesis and suggesting that Agrin and Lrp4 are likely to work together in the pathway.

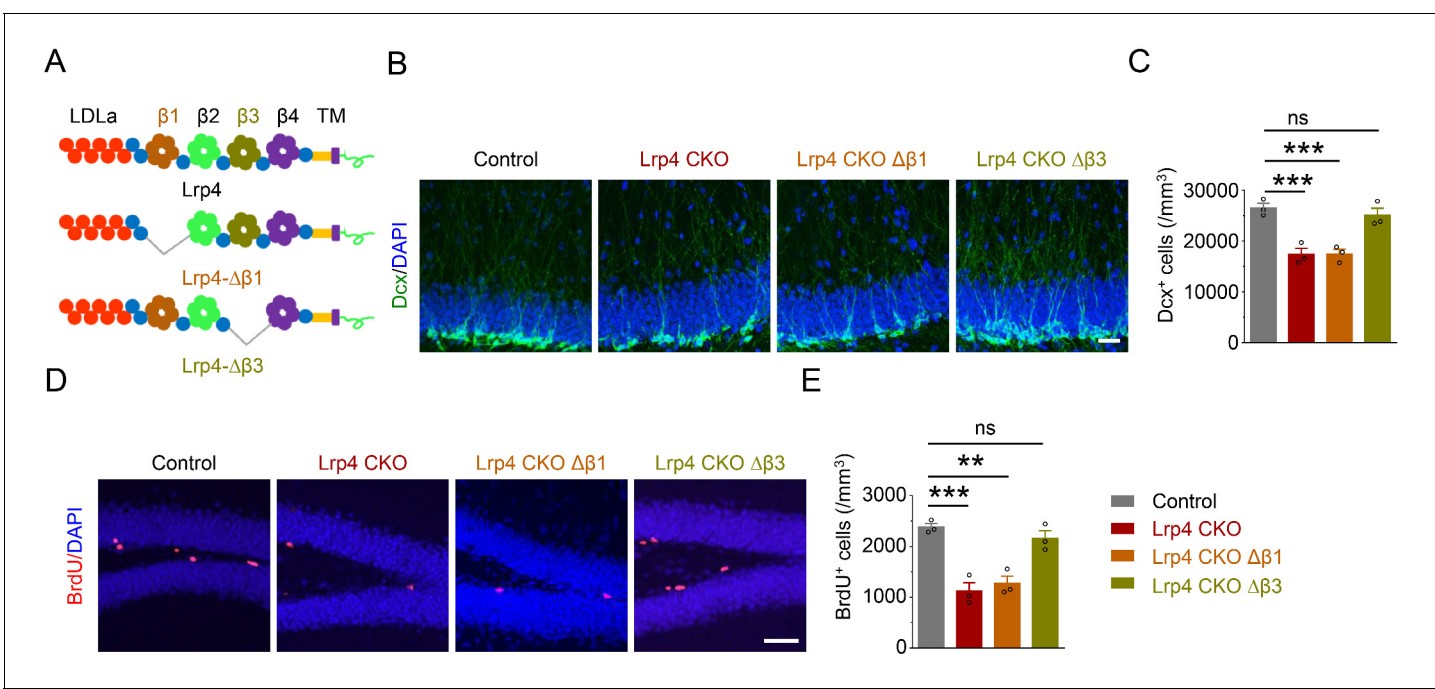

**Figure 5.** Requirement of the β1 propeller domain for Lrp4 regulation of adult neurogenesis. (**A**) Domain structures of Lrp4 and deletion mutants. (**B–C**) Reduced Dcx^+ cell density in Lrp4 CKO and Lrp4 CKO Δβ1, compared with control and Lrp4 CKO Δβ3 mice. (**B**) Representative images. Scale bar, 50 μm. (**C**) Stereological quantification of Dcx^+ cell density. n = 3 for each group. One-way ANOVA multiple comparisons test, F (3,8)=18.69, p=0.0006; control vs Lrp4 CKO, p=0.0011; control vs Lrp4 CKO Δβ1, p=0.0012; control vs Lrp4 CKO Δβ3, p=0.7005. (**D–E**) Reduced BrdU^+ cell density in Lrp4 CKO and Lrp4 CKO Δβ1, compared with control and Lrp4 CKO Δβ3 mice. (**D**) Representative images. Scale bar, 50 μm. (**E**) Stereological quantification. One-way ANOVA multiple comparisons test, F (3,8)=21.26, p=0.0004; control vs Lrp4 CKO, p=0.0005; control vs Lrp4 CKO Δβ1, p=0.0011; control vs Lrp4 CKO Δβ3, p=0.5538. Data are mean ± s.e.m. ns, p>0.05; **, p<0.01; ***, p<0.001.

DOI: https://doi.org/10.7554/eLife.45303.019

The following source data and figure supplement are available for figure 5:

**Source data 1.** Requirement of the β1 propeller domain for Lrp4 regulation of adult neurogenesis.

DOI: https://doi.org/10.7554/eLife.45303.021

**Figure supplement 1.** Generation of Lrp4 CKO Δβ1 and Lrp4 CKO Δβ3 mice.

DOI: https://doi.org/10.7554/eLife.45303.020

Intriguingly, the deficits were mitigated by expressing Lrp4Δβ3, indicating that the β3 domain is dispensable and suggesting the involvement of a receptor tyrosine kinase other than MuSK. Among receptor tyrosine kinases, Rors show the highest homology to MuSK (*Masiakowski and Yancopoulos, 1998*). There are two Ror kinases, Ror1 and Ror2, which were thought to be orphan receptors until recent evidence that they may in part function as receptors for Wnt5a (*Ho et al., 2012*; ; *Mikels et al., 2009*; *Oishi et al., 2003*). To investigate whether Rors play a role in Agrin-Lrp4 signaling, we first determined whether they interact with Lrp4. *Flag-Lrp4* and HA-tagged *Ror*1 and *Ror*2 were co-transfected into HEK293T cells. Lrp4 was precipitated from cells lysates by a Flag antibody, and the resulting immunocomplex was analyzed for HA-Ror1 and Ror2. As shown in *Figure 6A*, Ror2 coprecipitated with Lrp4 in transfected cells. This interaction appeared to be specific because Ror1 did not co-precipitate with Lrp4 from cell lysates (*Figure 6B*). These results support the notion that Ror2, but not Ror1, may serve as a downstream kinase of Lrp4. In support of this notion was the finding that the Lrp4-Ror2 interaction was enhanced by Agrin stimulation (*Figure 6C and D*).

To explore Ror2's function in adult neurogenesis, we cultured neurospheres from DG and stimulated them with Agrin. As shown in *Figure 6E*, Agrin increased the tyrosine-phosphorylation level of Ror2 in neurospheres. To determine whether Ror2 is necessary for adult neurogenesis in vivo, we generated *Ror*2 knockout mice by crossing *Ror2^{f/f}* mice with h*GFAP-Cre* mice (*Figure 6—figure supplement 1A*). Ror2 protein was reduced in the hippocampus of *GFAP-Cre::Ror2^{f/f}* (referred as Ror2 CKO) mice (*Figure 6—figure supplement 1B and C*). The brain size of Ror2 CKO mice was comparable to that of littermate control, and their hippocampal morphology appeared to be normal (*Figure 6—figure supplement 1D–1F*), in agreement with previous reports (*Endo et al., 2017*). We found that the density of Dcx^+ and BrdU^+ cells was reduced in Ror2 CKO mice, compared with control littermates (*Figure 6—figure supplement 1G–1J*), suggesting a necessary role of Ror2 in adult neurogenesis. Agrin-induced growth was blocked in neurosphere derived from Ror2 CKO or Lrp4 CKO mice (*Figure 6F and G*), indicating that the regulation by Ror2 and Lrp4 was likely to be cell-autonomous.

To test this hypothesis further, we generated *Nes-Cre/ERT2::Ror2^{f/f}* mice to knock out Ror2 specially in NSPCs by Tam injection (*Figure 6—figure supplement 2A–2D*). Dcx^+ cells are reduced in Tam-injected *Nes-Cre/ERT2::Ror2^{f/f}* (referred as Nes Ror2 CKO) mice, compared with Tam-injected *Nes-CreERT2::Ror2^{+/+}* mice (referred as control) mice (*Figure 6H and I*). Similar reduction was observed with BrdU^+ cells (*Figure 6J and K*). However, the density of NSCs (Gfap^+/Sox2^+/ BrdU^+) was similar between control and Nes Ror2 CKO mice (*Figure 6J and L*). These results support the notion that Ror2 in NSPCs is necessary for adult neurogenesis. Together, these results demonstrate indispensable roles of Lrp4 and Ror2 in adult neurogenesis and support a working model where Agrin binds to Lrp4 to activate Ror2 to promote adult neurogenesis in the hippocampus.

## Discussion

Adult hippocampal neurogenesis may be a mechanism for the brain to adapt to environmental changes. For example, it is increased in mice by exposure to EE, task learning, and physical exercise (*Faigle and Song, 2013*; *Gonçalves et al., 2016*). This dynamic, complex process is regulated by various factors (*Gonçalves et al., 2016*). Evidence indicates that Shh regulates NSCs self-renewal, proliferation, and migration (*Ahn and Joyner, 2005*; *Lai et al., 2003*); Notch signaling promotes cell cycle exit and decreases the neural progenitor pool; Wnt/beta-catenin signaling regulates NSCs proliferation and neuronal differentiation (*Lie et al., 2005*); and BMP signaling enhances glial differentiation (*Guo et al., 2011*; *Mira et al., 2010*). Tyrosine kinase activation by FGF-2, IGF-1, VEGF could stimulate the proliferation of NSPCs (*Faigle and Song, 2013*) whereas NT-3 and NGF regulate their differentiation or survival (*Frielingsdorf et al., 2007*; *Shimazu et al., 2006*). In addition, VEGF as well as BDNF signaling has been implicated in EE-enhanced hippocampal neurogenesis (*Cao et al., 2004*; *Fabel et al., 2003*; *Jin et al., 2002*; *Li et al., 2008*; *Rossi et al., 2006*).

We show here that *Agrn* is upregulated at the mRNA level in mouse hippocampus following EE exposure, consistent with a previous study that *Agrn* expression was activity-dependent (*O'Connor et al., 1995*). Mutation of *Agrn* in excitatory neurons decreases adult hippocampal neurogenesis, impaires the spatial memory and increases the immobility of mice in FST and TST. These results uncover a potentially novel function of Agrin. In NMJ formation, Agrin binds to Lrp4 to form

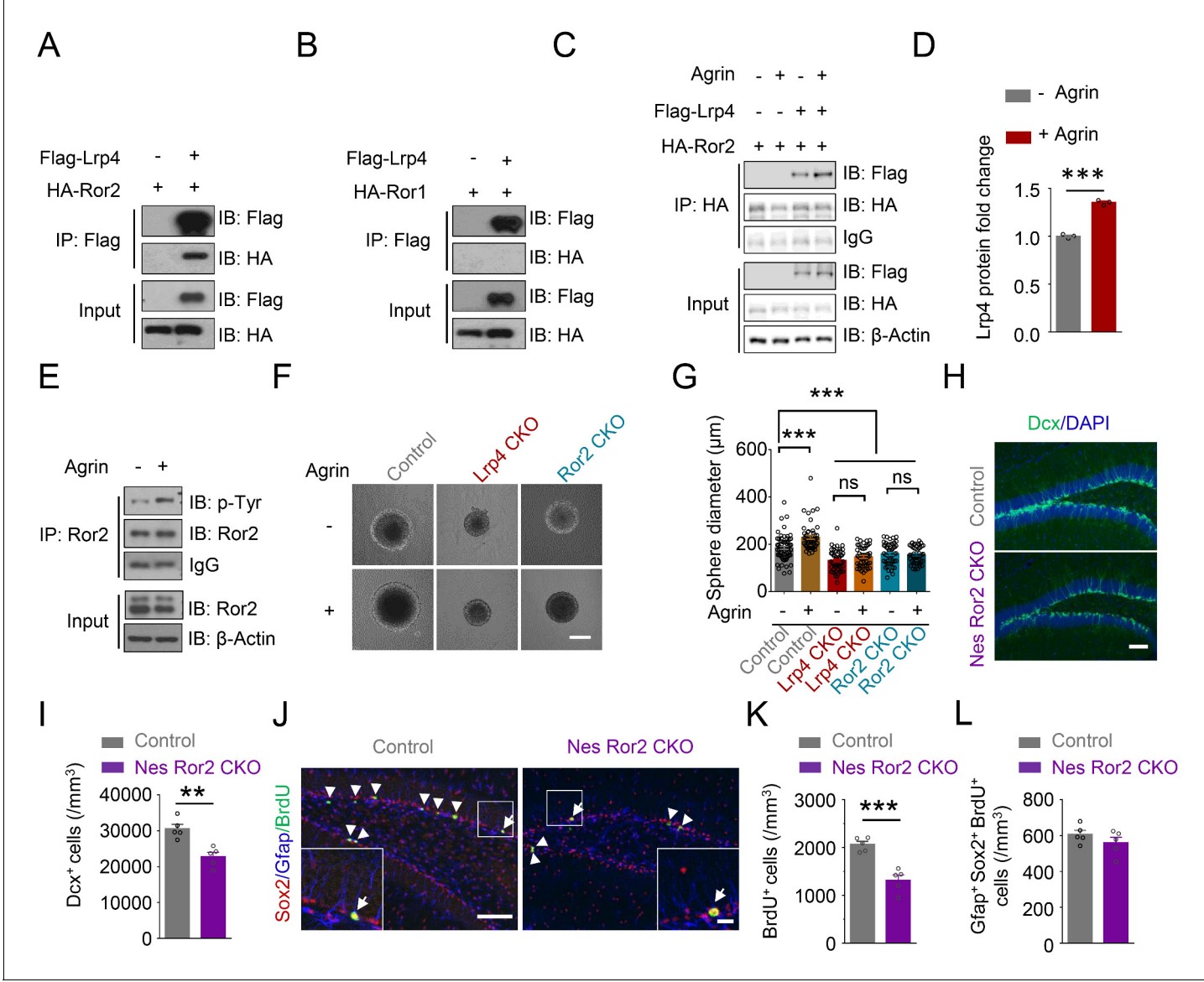

**Figure 6.** Requirement of Ror2 for adult neurogenesis. (**A–B**) Co-immunoprecipitation Ror2 (**A**), not Ror1 (**B**), with Lrp4 in co-transfected HEK293T cell. (**C–D**) Increased Lrp4-Ror2 interaction in Agrin-treated HEK293T cell. (**D**) Quantitative analysis of data of C. Lrp4 intensity was normalized by that of IgG. Student's t-test: t (4)=18.47, p<0.0001. (**E**) Increased Ror2 tyrosine phosphorylation in Agrin-treated neurosphere. Three independent experiments were performed. (**F–G**) Increased neurosphere size by Agrin and blockade by *Lrp4* or *Ror2* mutation. (**F**) Representative images. Scale bar, 100 μm. (**G**) Quantification of neurosphere size. One-way ANOVA test: F (5,314)=28.55, p<0.0001. Three independent experiments were performed. (**H–I**) Decreased Dcx[+] cell density in Nes Ror2 CKO mice, compared with control. (**H**) Representative images. Scale bar, 100 μm. (**I**) Stereological quantification of Dcx[+] cell density. n = 5 for each group. Student's t-test: t (8)=4.523, p=0.0019. (**J–L**) Reduced BrdU[+] cell density in Nes Ror2 CKO mice, compared with control. (**J**) Representative images. Scale bar 100 μm. (**K**) Stereological quantification of BrdU[+] cell density. n = 5 for each group. Student's t-test: t (8) =5.948, p=0.0003. (**L**) Similar density of SGZ Gfap[+]Sox2[+] BrdU[+] NSCs between the two genotypes. Student's t-test: t (8)=1.22, p=0.2572. Data are mean ± s.e.m. **, p<0.01; ***, p<0.001.

DOI: https://doi.org/10.7554/eLife.45303.022

The following source data and figure supplements are available for figure 6:

**Source data 1.** Requirement of Ror2 for adult neurogenesis.
DOI: https://doi.org/10.7554/eLife.45303.025
**Source data 2.** Characterization of Ror2 mutant mice.
DOI: https://doi.org/10.7554/eLife.45303.026
**Source data 3.** Characterization of inducible NSPCs-specific Ror2 knockout mice.
DOI: https://doi.org/10.7554/eLife.45303.027
*Figure 6 continued on next page*

*Figure 6 continued*

**Figure supplement 1.** Generation and characterization of *Ror2* mutant mice.
DOI: https://doi.org/10.7554/eLife.45303.023
**Figure supplement 2.** Generation and characterization of inducible NSPCs-specific *Ror2* knockout mice.
DOI: https://doi.org/10.7554/eLife.45303.024

an initial heterodimer, two of which form a tetrameric complex to activate the receptor tyrosine kinase MuSK (*Kim et al., 2008*; *Zhang et al., 2008*; *Zong et al., 2012*). Lrp4 is expressed in the brain, mostly concentrated in NSPCs and astrocytes (*Sun et al., 2016*). In particular, NSPCs-specific knockout of Lrp4 (by *Nes-CreERT*2) blocked EE-induced increase in BrdU$^+$ and Ki67$^+$ cells and reduction in mouse immobility. MuSK activation by Agrin/Lrp4 requires the β3 propeller domain (*Zhang et al., 2011*); yet a *Lrp*4 mutant lacking this domain was able to rescue adult neurogenesis deficits in Lrp4 CKO mice, suggesting that MuSK may not be involved. We show that Lrp4 interacts with Ror2, a transmembrane tyrosine kinase of the Ror family that is closely related to MuSK (*Green et al., 2014*; *Masiakowski and Yancopoulos, 1998*). There are two members in the Ror family: Ror1 and Ror2. Ror2 has an active kinase domain where Ror1 is inactive and may function as a pseudokinase (*Gentile et al., 2011*). In development, Ror1 and Ror2 act as receptor or coreceptor for Wnt5a to regulate cellular polarity, migration, proliferation, and differentiation via non-canonical Wnt pathway (*Green et al., 2014*) and maintain neuronal progenitor cell fate in the neocortex (*Endo et al., 2012*). Interestingly, Lrp4 interacts with Ror2, but not Ror1; and this interaction is enhanced by Agrin stimulation with concurrent increase in tyrosine phosphorylation. Remarkably, Agrin-induced proliferation of neural stem cells is attenuated by *Ror*2 mutation as well as *Lrp*4 mutation. *Ror*2 mutation, driven by *GFAP-Cre* or *Nes-CreERT*2, impairs adult neurogenesis. A parsimonious interpretation of these results is that upon activation, pyramidal neurons release Agrin, which binds to Lrp4 and activates Ror2 in NSPCs to promote adult neurogenesis.

Emerging evidence suggests a role for astrocytes in regulating adult neurogenesis, from proliferation and fate specification of neural progenitors to migration and integration of neural progeny into existing brain circuits (*Song et al., 2002*). Astrocytes are in intimate contact with NSPCs and produce both membrane-bound and soluble factors to stimulate NSPCs to reenter the cell cycle and adopt a neuronal fate (*Song et al., 2002*). Implicated factors include ATP, Wnt3, D-Serine, and thrombospondin (*Cao et al., 2013a*; *Lie et al., 2005*; *Lu and Kipnis, 2010*; *Sultan et al., 2015*). Interestingly, Lrp4 is highly expressed in astrocytes in various brain regions including the hippocampus (*Sun et al., 2016*). A function of Lrp4 in astrocytes is to control ATP release; *Lrp*4 mutation increases ATP in the brain and the condition medium of cultured astrocytes (*Sun et al., 2016*). However, the increase in ATP is unlikely to be a mechanism because astrocytic ATP has been shown to promote NPCs proliferation (*Cao et al., 2013a*; *Cao et al., 2013b*), a phenotype different from those of *Agrn* or *Lrp*4 mutant mice.

Our study demonstrates a cell-autonomous role of Lrp4 for NSPCs proliferation. In addition to astrocytes, Lrp4 is expressed in precursor cells including RGLs and progenitors, but not in more mature neuroblasts or dentate granule neurons (*Habib et al., 2016*; *Hochgerner et al., 2018*; *Shin et al., 2015*). Lrp4 is a member of the LDL receptor (LDLR) family, with an enormous extracellular region that consisting of a LDLa domain, 4 β-propeller domains and several EGF-like domains (*Herz, 2009*; *Shen et al., 2015*). Ligands of Lrp4, beside Agrin, include Dickkopf-1 (DKK1), Wnt, APP, ApoE, sclerostin, gremlin1, and Wise (*Shen et al., 2015*). Several of these ligands have been implicated in adult neurogenesis. For example, loss of DKK1 restores neurogenesis in old age (*Seib et al., 2013*). APP deficiency enhances the proliferation of progenitor cells (*Wang et al., 2014*), whereas ApoE deficiency stimulates astrogenesis and inhibits neurogenesis (*Li et al., 2009*). Future studies will be necessary to determine whether Lrp4 contributes to effects of these factors.

Exercise is known to increase cell proliferation in the hippocampus (*Choi et al., 2018*; *van Praag et al., 1999*). Most if not all EE paradigms include a running wheel (*Kempermann et al., 1997*; *Sztainberg and Chen, 2010*; *van Praag et al., 1999*). Whether EE alone has a similar effect was controversial. EE with one running wheel in a large cage (with 12–14 mice) did not affect cell proliferation (*Kempermann et al., 1997*; *van Praag et al., 1999*) or a potentiating effect in 129/SvJ mice (but not C57/B6 mice) (*Kempermann et al., 1998*). When the number of running wheels was increased, cell proliferation effect was observed in C57/B6 mice (*Kobilo et al., 2011*). While EE with

a running wheel increased BrdU$^+$ cells in dentate gyrus, but EE without it was unable to do so (*Kobilo et al., 2011*). Interestingly, the inclusion of a running wheel did not further increase the number of BrdU$^+$ cells that were increased by EE (*Kempermann, 2015*). Consistently, we showed here that cell proliferation was increased in control mice by EE with two running wheels. Among BrdU$^+$ cells could be RGLs (Gfap$^+$/Nestin$^+$ or Gfap$^+$/Sox2$^+$), intermediate progenitor cells (Tbr2$^+$), and neuroblasts (PSA-NCAM$^+$ or Dcx$^+$) (*Ming and Song, 2011*). Neither EE nor running has any effect on the number of RGLs (*Kronenberg et al., 2003*). However, running, but not EE, increases the proliferation of intermediate progenitor cells. EE seems to promote the survival of neuroblasts (*Kronenberg et al., 2003*). Because Lrp4 is enriched in RGLs and intermediated progenitor cells, the reduction of BrdU$^+$ cells by Lrp4 mutation is likely due to reduced proliferation of intermediate progenitor cells. RGLs seen to be heterogeneous population and different subpopulation display discrete proliferation responses to running (*DeCarolis et al., 2013*; *Gebara et al., 2016*). Whether Lrp4-expressing cells represent a subtype of intermediate progenitor cells warrant future study. Finally, deceased adult hippocampal neurogenesis appears to associate with depressive-like behavior (*Airan et al., 2007*; *Czéh et al., 2002*; *Santarelli et al., 2003*; *Snyder et al., 2011*) although this notion was debatable (*Anacker and Hen, 2017*). Antidepressant regiments could upregulate adult hippocampal neurogenesis (*Anacker and Hen, 2017*). Our study identifies a previously not appreciated Agrin pathway in adult neurogenesis that warrants further investigation.

# Materials and methods

## Key resources table

| Reagent type (species) or resource | Designation | Source or reference | Identifiers | Additional information |
|---|---|---|---|---|
| Genetic reagent (*M. musculus*) | *Agrn*$^f$ | Jackson Laboratory | Stock #: 031788 | *Harvey et al., 2007* |
| Genetic reagent (*M. musculus*) | *Lrp4*$^f$ | *Wu et al., 2012* | | |
| Genetic reagent (*M. musculus*) | *Ror2*$^f$ | Jackson Laboratory | Stock #: 018354 | *Ho et al., 2012* |
| Genetic reagent (*M. musculus*) | *Neurod6-Cre* | CARD R-BASE | CARD ID: 2556 | *Goebbels et al., 2006* |
| Genetic reagent (*M. musculus*) | *GFAP-Cre* | Jackson Laboratory | Stock #: 004600 | *Zhuo et al., 2001* |
| Genetic reagent (*M. musculus*) | *Ai9* (B6.Cg-Gt(ROSA)26So$^{rtm9(CAG-tdTomato)Hze}$/J) | Jackson Laboratory | Stock #: 007909 | *Madisen et al., 2010* |
| Genetic reagent (*M. musculus*) | *Nes-Cre/ERT2* (C57BL/6Tg(Nes-cre /ERT2)KEisc/J) | Jackson Laboratory | Stock #: 016261 | PMID:17166924 |
| Genetic reagent (*M. musculus*) | *Lrp4-LacZ* | KNOCKOUT MOUSE PROJECT | Project ID: VG15248 | *Sun et al., 2018* |
| Genetic reagent (*M. musculus*) | *LSL-Lrp4-Δβ1* | This paper | | |
| Genetic reagent (*M. musculus*) | *LSL-Lrp4-Δβ3* | This paper | | |
| Cell line (*Homo sapiens*) | HEK293T | ATCC | Cat#:CRL-3216 RRID: CVCL_0042 | |
| Cell line (*Homo sapiens*) | GP2-293 | Clontech | Cat #: 631458 RRID: CVCL_WI48 | |
| Antibody | Mouse anti-Arc | Santa Cruz Biotechnology | Cat #: sc-7839 RRID: AB_626696 | IHC (1:200) |
| Antibody | Goat anti-Dcx | Santa Cruz Biotechnology | Cat #: sc-8066 RRID: AB_2088494 | IHC (1:200) |

*Continued on next page*

*Continued*

| Reagent type (species) or resource | Designation | Source or reference | Identifiers | Additional information |
|---|---|---|---|---|
| Antibody | Mouse anti-Mcm2 | BD Biosciences | Cat #: 610701 RRID: AB_398024, | IHC (1:500) |
| Antibody | Rat anti-BrdU | Accurate Chemical and Scientific Corporation | Cat #: OBT0030 RRID: AB_2313756 | IHC (1:500) |
| Antibody | Rabbit anti-Ki67 | Millipore | Cat #: AB9260 RRID: AB_2142366 | IHC (1:200) |
| Antibody | Mouse anti-Nestin | BD Biosciences | Cat #: 556309 RRID: AB_396354 | IHC (1:200) |
| Antibody | Rabbit anti-GFAP | Dako | Cat #: Z0334 RRID: AB_10013382 | IHC (1:1000) |
| Antibody | Chicken anti-β-gal | Aves Labs | Cat #: BGL-1040 RRID: AB_2313507 | IHC (1:1000) |
| Antibody | Mouse anti-Sox2 | Santa Cruz Biotechnology | Cat #: sc-20088 RRID: AB_2255358 | IHC (1:200) |
| Antibody | Rabbit anti-Tbr2 | Abcam | Cat #: ab23345 RRID: AB_778267 | IHC (1:1000) |
| Antibody | Mouse anti-PSA-NCAM | Millipore | Cat #: MAB5324 RRID: AB_95211 | IHC (1:500) |
| Antibody | Rabbit anti-Cleaved Caspase3 | Cell Signaling Technology | Cat #: 9661 RRID: AB_2341188 | IHC (1:200) |
| Antibody | Mouse anti-NeuN | Millipore | Cat #: MAB377 RRID: AB_2298772 | IHC (1:1000) |
| Antibody | Chicken anti-GFP | AVES | Cat #: GFP-1020 RRID: AB_10000240 | IHC (1:1000) |
| Antibody | Rabbit anti-Flag | Sigma-Aldrich | Cat #: F7425 RRID: AB_439687 | WB (1:1000) |
| Antibody | Mouse anti-HA | Sigma-Aldrich | Cat #: H9658 RRID: AB_260092 | WB (1:5000) |
| Antibody | Mouse anti-GAPDH | Santa Cruz Biotechnology | Cat #: sc-32233, RRID: AB_627679 | WB (1:10000) |
| Antibody | Mouse anti-β-Actin | Cell Signaling Technology | Cat #: 12262 RRID: AB_2566811 | WB (1:5000) |
| Antibody | Mouse anti-P-Tyr-100 | Cell Signaling Technology | Cat #: 9411 RRID: AB_331228 | WB (1:1000) |
| Antibody | Rabbit anti-Ror2 | Cell Signaling Technology | Cat #: 4105 RRID: AB_2180134 | WB (1:1000) |
| Antibody | Mouse anti-Lrp4 | UC Davis/NIH NeuroMab Facility | Cat #: 75–221 RRID: AB_2139030 | WB (1:1000) |
| Antibody | Alexa Fluor 647-AffiniPure Fab Fragment Donkey Anti-Rabbit IgG (H + L) | Jackson Immuno Research Labs | Cat #: 711-607-003 RRID: AB_2340626 | IHC (1:200) |
| Antibody | Alexa Fluor 594-AffiniPure F(ab')2 Fragment Donkey Anti-Rabbit IgG (H + L) | Jackson Immuno Research Labs | Cat #: 711-586-152 RRID: AB_2340622 | IHC (1:200) |
| Antibody | Alexa Fluor 488-AffiniPure Fab Fragment Donkey Anti-Rabbit IgG (H + L) | Jackson Immuno Research Labs | Cat #: 711-547-003 RRID: AB_2340620 | IHC (1:200) |

*Continued*

| Reagent type (species) or resource | Designation | Source or reference | Identifiers | Additional information |
|---|---|---|---|---|
| Antibody | Alexa Fluor 647-AffiniPure Fab Fragment Donkey Anti-Mouse IgG (H + L) | Jackson Immuno Research Labs | Cat #: 715-607-003 RRID: AB_2340867 | IHC (1:200) |
| Antibody | Alexa Fluor 488-AffiniPure Fab Fragment Donkey Anti-Mouse IgG (H + L) | Jackson Immuno Research Labs | Cat #: 715-547-003 RRID: AB_2340851 | IHC (1:200) |
| Antibody | Alexa Fluor 488-AffiniPure Fab Fragment Donkey Anti-Goat IgG (H + L) | Jackson Immuno Research Labs | Cat #: 705-547-003 RRID: AB_2340431 | IHC (1:200) |
| Antibody | Alexa Fluor 488-AffiniPure F(ab')2 Fragment Donkey Anti-Chicken IgY (IgG) (H + L) | Jackson Immuno Research Labs | Cat #: 703-546-155 RRID: AB_2340376 | IHC (1:200) |
| Antibody | Alexa Fluor 647-AffiniPure Fab Fragment Donkey Anti-Rat IgG (H + L) | Jackson Immuno Research Labs | Cat #: 712-607-003, RRID: AB_2340697 | IHC (1:200) |
| Antibody | IRDye 680RD Donkey anti-Rabbit IgG (H + L) | LI-COR Biosciences | Cat #: 926–68073, RRID: AB_10954442 | WB (1:10000) |
| Antibody | Donkey Anti-Mouse IgG, IRDye 800CW Conjugated | LI-COR Biosciences | Cat # 926–32212, RRID: AB_621847 | WB (1:10000) |
| Recombinant DNA reagent | pFlag-Lrp4 | PMID: 30171091 | | Materials and methods subsection: antibodies and plasmid |
| Recombinant DNA reagent | HA-Ror1 | This paper | | Materials and methods subsection: antibodies and plasmid |
| Recombinant DNA reagent | HA-Ror2 | This paper | | Materials and methods subsection: antibodies and plasmid |
| Chemical compound, drug | BrdU | Sigma | Cat #: B5002 | |
| Chemical compound, drug | Tamoxifen | Sigma | Cat #: T5648 | |
| Software, algorithm | Image J | NIH, USA | RRID:SCR_003070 | |

## Animals

The following mice were described previously: *Agrn*[f] (*Harvey et al., 2007*), *Lrp4*[f] (*Wu et al., 2012*), *GFAP-Cre* (*Zhuo et al., 2001*), *Ai9* (*Madisen et al., 2010*) (Jackson Labs, #007909), *Nes-Cre/ERT2* (Jackson Labs, #016261), *Lrp4-LacZ* reporter mice were from UCDAVIS KOMP Respository (VG15248) (*Sun et al., 2016*), *Ror2*[f] (*Ho et al., 2012*), *Neurod6-Cre* (*Goebbels et al., 2006*). *LSL-Δβ* 1 and *LSL-Δβ*3 transgenic mice were generated by subcloning respective insert (Lrp4 without aa 435–749 and Lrp4 without aa 1045–1354) into pCCALL2 at *Hind* III and *Not* I sites, which was confirmed by sequencing. The transgenes were purified from vector sequences and microinjected into the pronuclei of single-cell C57BL/6JxSJL hybrid embryos. Founder transgenic mice were identified by PCR. Primers for genotypes were as follow: LSL-Δβ1 (F: 5' CCA GGA TGT GAA TGA ATG TG 3',

R: 5' ACT TGT CGG TTG GAG GC 3'); LSL-Δβ3 (F: 5' ACA CGG ACG GCA GCA T 3', R: 5' AGC CCA TCA GTG GTC TTC 3'). Mice were group-housed no more than five per cage in a room with a 12-h light/dark cycle with ad libitum access to water and rodent chow diet (Diet 1/4 7097, Harlan Teklad). In some experiments, mice were housed for 4 weeks in EE cages (86 cm x 76 cm x 24 cm; l x w x h; 12 mice per cage) with regular bedding, food and water ad libitum, and EE items (two running wheels with solid closed plastic floor, two plastic tubes, one red transparent plastic nest box and a paper-based nest box). Experiments with animals were approved by the Institutional Animal Care and Use Committee of Augusta University and Case Western Reserve University. Male mice were used for all the studies.

## Antibodies and plasmids

The information of primary antibodies used was as follows: mouse anti-Arc (Santa Cruz Biotechnology, sc-7839); goat anti-Dcx (Santa Cruz Biotechnology, sc-8066); mouse anti-Mcm2 (BD Biosciences, 61070); rat anti-BrdU (Accurate Chemical and Scientific Corporation, OBT0030); rabbit anti-Ki67 (Millipore, AB9260); mouse anti-Nestin (BD Biosciences, 556309); rabbit anti-GFAP (Dako, Z0334); chicken anti-β-gal (Aves Labs, BGL-1040); mouse anti-Sox2 (Santa Cruz Biotechnology, sc-20088); rabbit anti-Tbr2 (Abcam, ab23345); mouse anti-PSA-NCAM (Millipore, MAB5324); rabbit anti-Cleaved Caspase-3 (Cell Signaling Technology, 9661); mouse anti-NeuN (Millipore, MAB377); chicken anti-GFP (AVES, GFP-1020); rabbit anti-Flag (Sigma-Aldrich, F7425); mouse anti-HA (Sigma-Aldrich,H9658); mouse anti-GAPDH (Santa Cruz Biotechnology, sc-32233); mouse anti-β-Actin (Cell Signaling Technology, 12262); mouse anti-P-Tyr (Cell Signaling Technology, 9411); rabbit anti-Ror2 (Cell Signaling Technology, 4105); and mouse anti-Lrp4 (UC Davis/NIH NeuroMab Facility, 75–221). The information of secondary antibodies used was as follows: Alexa Fluor 488-donkey-anti-mouse-IgG (Cat #: 715-547-003); Alexa Fluor 647-donkey-anti-mouse-IgG (Cat #: 715-607-003); Alexa Fluor 594-donkey-anti-rabbit-IgG (Cat.#711-586-152); Alexa Fluor 488-donkey-anti-rabbit-IgG (Cat #: 711-547-003); Alexa Fluor 647-donkey-anti-rabbit-IgG (Cat #: 711-607-003); Alexa Fluor 647-donkey-anti-mouse-IgG (Cat #: 715-607-003); Alexa Fluor 488-donkey-anti-goat-IgG (Cat #: 705-547-003); Alexa Fluor 488-donkey-anti-chicken-IgG (Cat #: 703-546-155); Alexa Fluor 594-donkey-anti-rat-IgG (Cat #: 712-607-003), all 1:200 (Jackson ImmunoResearch Laboratories Inc, West Grove, PA) for IHC. RDye680RD Donkey anti-Rabbit-IgG (H + L, LI-COR Bioscience, Cat #: P/N 926–68073) and IRDye 800CW Donkey anti-Mouse-IgG (H + L, LI-COR Bioscience, Cat #: P/N 926–32212) secondary antibodies, both 1:10000 were used for western blot. D-Cre-GFP and Cre-GFP plasmids were a kind gift from Dr. Weixiang Guo. Lrp4 Rat cDNA were generated by PCR and subcloned into pFLAG-CMV1 (Sigma, Cat # E7273). Ror1 and Ror2 mouse cDNA were generated by PCR and subcloned into pKH3 (Addgene, RRID: Addgene_12555). Authenticity of all constructs was verified by DNA sequencing.

## 5-Bromo-2'-deoxyuridine (BrdU) and Tamoxifen administration

Mice were injected with BrdU and Tamoxifen as previously described (Appel et al., 2018). Briefly, mice were injected with BrdU (Sigma, 10 mg/mL, B5002; i.p., 200 mg/kg body weight) 2 h before perfusion. Tamoxifen (10 mg/mL, Sigma, T5648) was prepared in corn oil (Sigma, C8267) mixed with ethanol (9:1 ratio). Mice were injected with 100 mg/kg Tamoxifen (i.p., daily for constitutive 5 days) for 4-week-old male mice and with 125 mg/kg Tamoxifen (i.p., every 12 h for 4 times) for 8-week-old male mice. Mice were perfused at 1 month after injection and 2 days after injection, respectively.

## In situ X-gal assay

In situ X-gal assay was carried out as previously described (Sun et al., 2016). Briefly, brains were quickly isolated and embedded in OCT (Tissue-Tek). Coronal sections were cut at 20 µm in thickness, and every fourth section was collected and mounted onto slides. Sections were fixed for 2 min in PBS containing 2 mM MgCl2 and 5 mM EGTA with 0.2% glutaraldehyde. Sections were washed in ice-cold PBS and stained in X-gal solution (1 mg/mL X-gal, 5 mM K3Fe(CN)6, 5 mM K4Fe(CN)6, 0.02% NP-40, 0.01% deoxycholate, and 2 mM MgCl2 in PBS) at 37°C overnight. Following a wash with PBS, sections were counterstained with nuclear Fast Red (Vector Labs, H-3403).

## Immunostaining

Immunostaining was performed as described previously (*Sun et al., 2016*). Briefly, mice were deeply anesthetized with isoflurane and perfused with PBS followed by 4% paraformaldehyde (PFA) until bodies became stiff. Brain was post-fixed in 4% PFA at 4°C for another 8 h and dehydrated using 30% sucrose at 4°C for 2 days. Brain was embedded in optimal cutting temperature compound (4583; Tissue-Tek), rapidly frozen. Serial 40-μm-thick coronal brain sections were cut on a cryostat (HM550; Thermo Scientific). Sections were permeabilized with 0.3% Triton X-100, blocked with 10% donkey serum for 1 h at room temperature, incubated with primary antibodies at 4°C overnight. After washing with PBS 3 times, incubated with corresponding conjugated secondary antibody for 2 h. DAPI was used for nucleus counterstaining.

## Stereological quantification

Stereological quantification of cells was carried out as previously described with a slight modification (*Appel et al., 2018*). Briefly, cells were counted in a one-in-six series of sections through hippocampus (Bregma -1.06 mm to -3.08 mm). DAPI staining was used to outline DG area using Image J software. The total number of marker$^+$ cells was counted, and the volume of the DG section was calculated by multiplying the area by its thickness. The cell count was divided by the resultant section volume to obtain the total cell density in the dentate gyrus per mm$^3$. The hippocampus volume was estimated by using a one-in-six systematic random series of 40 μm Nissl-stained brain sections. Image J software was used to outline and measure the hippocampus area. The total volume of hippocampus was estimated by multiplying the area with its thickness and the Cavalieri's principle. The investigator blind to the genotype.

## Quantitative real time-polymerase chain reaction (qRT-PCR)

Different brain regions were dissected and frozen in liquid nitrogen. Total RNA was purified using TRIzol (15596–026, Invitrogen). Total RNA (3 μg) was reverse-transcribed to cDNA (Promega) and subjected to qPCR using SYBR green (Qiagen) in CFX96 real-time system (Bio-Rad). Primer sequences used were as follows: *Lrp*4 (F: 5'GTG TGG CAG AAC CTT GAC AGTC 3', R: 5' TAC GGT CTG AGC CAT CCA TTC C 3'); *ApoE* (F: 5' GAA CCG CTT CTG GGA TTA CC TG 3', R: 5' GC CTT TAC TTC CGT CAT AGT GTC 3'); *Wnt5a* (F: 5' GGA ACG AAT CCA CGC TAA GGG T 3', R: 5' AGC ACG TCT TGA GGC TAC AGG A 3'); *Bdnf* (F: 5' GGC TGA CAC TTT TGA GCA CGT C-3', R: 5' CTC CAA AGG CAC TTG ACT GCT G-3'); *Igf*1 (F: 5' GTG GAT GCT CTT CAG TTC GTG TG 3', R: 5' TCC AGT CTC CTC AGA TCA CAG C 3'); *Vegf* (F: 5' CTG CTG TAA CGA TGA AGC CCT G 3', R: 5' GCT GTA GGA AGC TCA TCT CTC C 3'); *MuSK* (F: 5' CTG AAG GCT GTG AGT CCA CTG T 3', R: 5' TCC TTT ACC GCC AGG CAG TAC T 3'); *Agrn* (F: 5' AGA TGG TGT TCT TGG CTC GTG G 3', R: 5' CAG GGC TAT GGG CTC TTT GCT 3'); n*Agrn* (F: 5'CAC TGC GAG AAG GGG ATA GTT G-3', R: 5' GGC TGG GAT CTC ATT GGT CAG 3'); *GAPDH* (F: 5' CAT CAC TGC CAC CCA GA AGA CTG 3', R: 5' ATG CCA GTG AGC TTC CCG TTC AG 3'). Each sample was assayed in triplicate, and the mRNA level was normalized to GAPDH using the $2^{-\Delta\Delta CT}$ method.

## Depressive-like behavior test

Behavioral testing was performed during the light phase of the cycle, that is between 9:00 A.M. and 5:00 P.M. Mice (7–8 weeks) were habituated to test room for 3 days before forced swim test (FST), which was followed by 2 days of habituation and then tail suspension test (TST). A short habituation (2 h) was allowed on test day. FST and TST were carried out as previously described (*Appel et al., 2018*). Briefly, in FST, mice were individually placed in a glass cylinder (25 cm height, 10 cm diameter) with water (22°C). Mice were allowed to swim in water for 6 min and scored for immobile time in last 4 min. In TST, mice were individually suspended by the distal portion of tails with adhesive tape for 6 min and scored for immobile time in last 4 min. Tests were performed by investigators blind to genotypes.

## Morris water maze

The Morris water maze was performed as previously described with slight modification (*Sun et al., 2016*). A 120 cm pool and 10 cm platform were used for water maze and nontoxic bright white gel (Soft Gel Paste Food Color, AmeriColor) was added to the water to make the surface opaque and to

hide the escape platform (1 cm below the surface). Mice were trained for 5 days with four trials per day with 20 min interval between trials and 60 sec per trial to locate the hidden platform. Eight spatial cues on the pool wall are visible for mice to find the hidden platform. On the 6th day the platform was removed and mice were placed into the pool at new start position and assessed the time spent in the platform quadrant and the number of platform crossing within 60 sec. The swim speed and amount of time spend in each quadrant were quantified using the video tracking system (Noldus). The investigator was blind to genotype during the data acquisition and analysis.

### Object location test

The object location test was carried out as previously described with modifications (*Hattiangady et al., 2014*). Briefly, mice were habituated in the open field chamber (50 × 50 cm) for 10 min 24 h before starting the test. On the test day, mice were placed in the chamber with two identical objects for 10 min and returned to its home cage for 24 h. They were allowed to explore the two identical objects except one of them was placed to new location. The time that mice sniffed the objects were recorded and preference scores were calculated. The investigator was blind to genotype during the data acquisition and analysis.

### In vivo genetic manipulation of neural progenitors

Cre-GFP ($5.2 \times 10^7$ pfu/mL) and D-Cre-GFP ($4.6 \times 10^7$ pfu/mL) retroviruses were produced following the GP2-293 (RRID: CVCL_WI48) cells manual, which were purchased from Clontech and were certified authentic and found to be free of *Mycoplasma*. The $Lrp4^{f/f}$ mice (7–8 weeks old) were anesthetized and stereotaxically injected with a virus into DG (0.5 µL at 0.25 µL/min) with the following coordinates (posterior = -2.0 mm from Bregma, lateral = ± 1.6 mm, ventral = 2.0 mm) as previously described (*Zhang et al., 2016*). After perfusion with PBS and PFA as described above, coronal sections (50 µm) were prepared and processed for morphological analysis.

For analysis of dendrite development, three-dimensional (3D) reconstructed images of entire dendritic processes of individual GFP$^+$ neurons were obtained from Z-series stacks of confocal images. Two-dimensional (2D) projection images were traced with NIH Image J using the neuron J plugin. GFP$^+$ dentate granule cells with intact dendritic trees were analyzed for total dendritic length and complexity as previously described (*Zhang et al., 2016*). The measurements did not include corrections for inclinations of the dendritic process and therefore represented projected lengths. Images of GFP labeled dendritic processes at the outer molecular layer were acquired at 0.18 µm intervals with Zeiss LSM 800 Airyscan system with a plane apochromatic 63 x oil lens [numerical aperture (NA), 1.4; Zeiss] and a digital zoom of 3.2. The Zeiss image files were subjected to the Airyscan processing. The structure of dendritic fragments and spines was traced using 3D Imaris software using a 'fire' heat map and a 2D X-Y ortho slice plane to aid visualization (Bitplane). Dendritic processes were traced using automatic filament tracer, whereas dendritic spines were traced using an auto-path method with the semi-automatic filament tracer (diameter; min: 0.1, max: 2.0, contrast: 0.8) (*Zhang et al., 2016*). The spine density was calculated by dividing the total number of spines by the length of the dendritic segment. The investigator was blind to genotype during the image acquisition and analysis of data.

### Neurosphere assay

Neurospheres were prepared as described previously (*Sun et al., 2018*). Briefly, DG regions were isolated, minced and treated with papain (0.8 mg/mL) for 30 min at 37°C. Tissues were then mechanically dissociated in HBSS containing 30 mM glucose, 2 mM HEPES and 26 mM NaHCO3 to obtain single-cell suspension. Cells were seeded at a density of 5,000–10,000 cells/mL and cultured in culture medium containing Neural Basal Medium, 2% B27,1x GlutaMAX, 2 µg/mL heparin, 50 units/mL Penicillin/Streptomycin, 20 ng/mL epidermal growth factor, and 20 ng/mL fibroblast growth factor for 7 days. Neurospheres were treated without or with Agrin (100 ng/mL) for 7 days in culture medium and scored for size/diameter using Image J (NIH).

### Cell culture, transfection, co-immunoprecipitation, and western blotting

HEK293T cells were purchased from ATCC (RRID: CVCL_0063) and were certified authentic and found to be free of *Mycoplasma*. Cells were cultured in DMEM (Hyclone) supplemented with 10%

fetalbovine serum (FBS) and transfected with polyethyleneimine (PEI), as previously described (*Zhang et al., 2008*). Flag-tagged Lrp4 and Ror2 were immunoprecipitated with Flag M2 beads (SigmaA2220) and anti-Ror2 antibodies. Western blotting was performed as described previously (*Wang et al., 2018*). Three independent experiments were performed.

## Statistical analysis

Data are mean ± standard error of the mean (s.e.m.). For two independent data comparisons, unpaired student's t-test was used to determine statistical significance. For multiple comparisons, ANOVA was used. *, $p < 0.05$; **, $p < 0.01$; ***, $p < 0.001$. Statistical analyses were performed using Excel 2016 (Microsoft) or GraphPad Prism 6.0.

## Acknowledgements

We are grateful to Dr. Weixiang Guo (Institute of Genetics and Developmental Biology, Chinese Academy of Sciences) for *Cre-GFP* and *D-Cre-GFP* plasmids; Dr. Quansheng Du (Augusta University) for GP2-293 cell line; Dr. Eleni Markakis (Case Western Reserve University) for commenting on an early version of the manuscript; Dr. Kexin Jiao (Case Western Reserve University) for helping 3D reconstruction, and members of the Mei and Xiong Lab for suggestion on the manuscript. This work was supported in part by grants from the National Institutes of Health (MH083317, NS082007, NS090083, and AG051510 to LM; AG051773 and AG045781 to W-CX) and Veteran Administration Office and Development (1/01IBX001020A to LM).

## Additional information

### Funding

| Funder | Grant reference number | Author |
|---|---|---|
| National Institutes of Health | MH083317 | Lin Mei |
| National Institutes of Health | NS082007 | Lin Mei |
| National Institutes of Health | NS090083 | Lin Mei |
| National Institutes of Health | AG051510 | Lin Mei |
| National Institutes of Health | AG051773 | Wen-Cheng Xiong |
| National Institutes of Health | AG045781 | Wen-Cheng Xiong |
| Veterans Health Administration Office of Research and Development | 1/01IBX001020A | Lin Mei |

The funders had no role in study design, data collection and interpretation, or the decision to submit the work for publication.

### Author contributions

Hongsheng Zhang, Conceptualization, Resources, Data curation, Software, Formal analysis, Validation, Investigation, Visualization, Methodology, Writing—original draft, Project administration, Writing—review and editing; Anupama Sathyamurthy, Conceptualization, Resources, Data curation, Formal analysis, Investigation, Methodology; Fang Liu, Lei Li, Lei Zhang, Zhaoqi Dong, Wanpeng Cui, Xiangdong Sun, Kai Zhao, Hongsheng Wang, Resources, Methodology; Hsin-Yi Henry Ho, Resources; Wen-Cheng Xiong, Conceptualization, Resources, Formal analysis, Supervision, Funding acquisition; Lin Mei, Conceptualization, Resources, Data curation, Formal analysis, Supervision, Funding acquisition, Methodology, Writing—original draft, Project administration, Writing—review and editing

### Author ORCIDs

Hongsheng Zhang (iD) https://orcid.org/0000-0001-8138-2108
Hsin-Yi Henry Ho (iD) http://orcid.org/0000-0002-8780-7864

Wen-Cheng Xiong http://orcid.org/0000-0001-9071-7598
Lin Mei https://orcid.org/0000-0001-5772-1229

### Ethics

Animal experimentation: All procedures involving animals were in accordance with the National Institutes of Health Guide for the care and use of Laboratory Animals and approved by Institutional Animal Care and Use Committees of Augusta University (Protocol #: 2011-0393) and Case Western Reserve University (Protocol #: 2017-0115).

### Decision letter and Author response

Decision letter https://doi.org/10.7554/eLife.45303.030
Author response https://doi.org/10.7554/eLife.45303.031

## Additional files

### Supplementary files

• Transparent reporting form
DOI: https://doi.org/10.7554/eLife.45303.028

### Data availability

All data generated or analyzed during this study are included in the manuscript and supporting files.

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
