## [Decision Letter]

Thank you for submitting your article "Agrin-Lrp4-Ror2 signaling regulates adult hippocampal neurogenesis in mice" for consideration by *eLife*. Your article has been reviewed by two peer reviewers, one of whom is a member of our Board of Reviewing Editors, and the evaluation has been overseen by Marianne Bronner as the Senior Editor. The reviewers have opted to remain anonymous.

The reviewers have discussed the reviews with one another and the Reviewing Editor has drafted this decision to help you prepare a revised submission.

Summary:

In this manuscript, entitled "Agrin-Lrp4-Ror2 signaling regulates adult hippocampal neurogenesis in mice", the authors investigate the role of agrin-Lrp4 ligand-receptor signaling in adult hippocampal neurogenesis under normal homeostatic conditions and with environmental enrichment. First, they show that environmental enrichment increases both agrin and Lrp4 in the hippocampus. Then they show that deletion of the agrin ligand reduces proliferation and neurogenesis in the dentate gyrus, and impairs hippocampus-dependent tasks. Then they show that the agrin receptor, Lrp4, is expressed in astrocytes and neural stem cells, as well as low expression in progenitors. Then the authors delete Lrp4 in neural stem cells using the GFAP-Cre line and the Nestin-CreER line and show that proliferation and neurogenesis are reduced, and the positive effect of environmental enrichment is blocked at the cellular and behavioral levels. The authors go on to show that deletion of Lrp4 in dividing progenitors alters dendritic morphology and spine density of newly generated neurons. Lastly, the authors provide both in vivo and in vitro evidence that the downstream signaling partner of agrin-Lrp4 is Ror2 using Lrp4 mutation, immunoprecipitation, neurosphere cultures, and deletion of Ror2.

The manuscript is clearly written and the experiments follow a logical progression. The authors were also quite thorough and convincing in their experiments determining the downstream signaling partners involved in this pathway. However, a major concern is the use of a non-inducible Cre line (GFAP-Cre) in experiments investigating adult neurogenesis, as there could be confounding developmental effects. The authors also claim the Lrp4 regulates adult neural stem cell proliferation, but this is never clearly shown in the data presented. This manuscript could be improved with additional experiments and textual changes as outlined below.

Essential revisions:

1) Because the GFAP-Cre line is not inducible, it has a broad impact on many stem cells during development, and subsequently impacts the glial and neuronal progeny of those stem cells in the adult brain. The authors begin the paper by using the GFAP-Cre line to delete Lrp4 throughout life and show that it affects adult neurogenesis. Then they do a more elegant experiment using the Nestin-CreER line to inducibly delete Lrp4 in the adult brain and show similar reductions in adult neurogenesis. However, the authors go back to using the GFAP-Cre line for the remainder of the manuscript, which investigates the downstream signaling partner of Lrp4, Ror2. To truly study the impact of a protein on adult neurogenesis, it is necessary to use a technology that does not impact development, so as not to confound the results. It is unclear why the authors continued to use the GFAP-Cre line after they show that Lrp4 deletion using the Nestin-CreER line decreases neurogenesis. Ideally, the experiments in Figure 6 and 7 would be done using the Nestin-CreER line in place of the GFAP-Cre line. However, the experiments in Figure 6 would require an incredible amount of mouse breeding to complete in a reasonable amount of time. I would suggest doing some, if not all, of the same experiments in Figure 7F-K using the Nestin-CreER line.

2) The authors claim that Lrp4 is expressed in the adult neural stem cells of the dentate gyrus, but it is unclear from their experiments how Lrp4 knockout affects the neural stem cells.

• In Figure 3E the authors show that Lrp4 knockout does not change the number of GFAP^+^*Sox2*^+^ neural stem cells but write in the manuscript that "Lrp4 mutation reduced the number of BrdU^+^*Sox2*^+^ cells (Figure 3F)". However, this is not what they actually show – they show that Lrp4 knockout changes the percentage of BrdU^+^ cells that are *Sox2*^+^. This could be due to changes in the composition of the dividing cell population. The authors should instead quantify the number of GFAP^+^*Sox2*^+^ neural stem cells that are BrdU^+^ or MCM2+ to see if Lrp4 knockout affects neural stem cell division.

• In Figure 4, the authors show that overall proliferation is reduced by conditional Lrp4 knockout in Nestin+ cells, but they do not show whether Lrp4 knockout changes the number or proliferation of neural stem cells in the dentate gyrus.

If the authors find that Lrp4 knockout does not change the number or proliferation of the adult neural stem cells, then it is more likely that the effect of Lrp4 is at the progenitor level, rather than the stem cell level.

3) The authors show in Figure 4C that there are significantly fewer labeled cells in iNestin-Lrp4f/f only 2 days after tamoxifen administration, but they do not give any explanation for this effect. This effect occurs in a strikingly small amount of time, so it suggests that there is either a proliferation defect in the rapidly amplifying progenitors or a cell death effect. The authors should address this issue at a minimum in the text, and potentially with additional experiments.

---

## [Author Response]

Essential revisions:1) Because the GFAP-Cre line is not inducible, it has a broad impact on many stem cells during development, and subsequently impacts the glial and neuronal progeny of those stem cells in the adult brain. The authors begin the paper by using the GFAP-Cre line to delete Lrp4 throughout life and show that it affects adult neurogenesis. Then they do a more elegant experiment using the Nestin-CreER line to inducibly delete Lrp4 in the adult brain and show similar reductions in adult neurogenesis. However, the authors go back to using the GFAP-Cre line for the remainder of the manuscript, which investigates the downstream signaling partner of Lrp4, Ror2. To truly study the impact of a protein on adult neurogenesis, it is necessary to use a technology that does not impact development, so as not to confound the results. It is unclear why the authors continued to use the GFAP-Cre line after they show that Lrp4 deletion using the Nestin-CreER line decreases neurogenesis. Ideally, the experiments in Figure 6 and 7 would be done using the Nestin-CreER line in place of the GFAP-Cre line. However, the experiments in Figure 6 would require an incredible amount of mouse breeding to complete in a reasonable amount of time. I would suggest doing some, if not all, of the same experiments in Figure 7F-K using the Nestin-CreER line.

These points are well taken. As suggested, we generated *Nes-Cre/ERT2::Ror2^f/f^*mice and knocked out Ror2 specially in NSPCs by Tam injection (Figure 6—figure supplement 2). Dcx^+^ cells are reduced in Tam-injected *Nes-Cre/ERT*2::*Ror2^f/f^* (referred as Nes Ror2 CKO) mice, compared with Tam-injected *Nes-Cre/ERT*2::*Ror*2^+/+^mice (referred as control) mice (revised Figure 6H and 6I). Similar reduction was observed with BrdU^+^ cells (revised Figures 6J and 6K). However, the density of neural stem cells (Gfap^+^/*Sox2*^+^/BrdU^+^) was similar between control and Nes Ror2 CKO mice (revised Figure 6L). These results support the notion that Ror2 in NSPCs is necessary for adult neurogenesis.

2) The authors claim that Lrp4 is expressed in the adult neural stem cells of the dentate gyrus, but it is unclear from their experiments how Lrp4 knockout affects the neural stem cells.• In Figure 3E the authors show that Lrp4 knockout does not change the number of GFAP+Sox2^+^ neural stem cells but write in the manuscript that "Lrp4 mutation reduced the number of BrdU^+^Sox2^+^ cells (Figure 3F)". However, this is not what they actually show – they show that Lrp4 knockout changes the percentage of BrdU^+^ cells that are Sox2^+^. This could be due to changes in the composition of the dividing cell population. The authors should instead quantify the number of GFAP+Sox2^+^ neural stem cells that are BrdU^+^ or MCM2+ to see if Lrp4 knockout affects neural stem cell division.

Good suggestion. Data of newly performed experiments show that BrdU^+^Gfap^+^*Sox2*^+^ cells were similar between Lrp4 CKO and control mice (revised Figure 3D-F), suggesting that Lrp4 knockout may not affect stem cell division.

• In Figure 4, the authors show that overall proliferation is reduced by conditional Lrp4 knockout in Nestin+ cells, but they do not show whether Lrp4 knockout changes the number or proliferation of neural stem cells in the dentate gyrus.If the authors find that Lrp4 knockout does not change the number or proliferation of the adult neural stem cells, then it is more likely that the effect of Lrp4 is at the progenitor level, rather than the stem cell level.

Good point. As suggested, we quantified BrdU^+^ cells among Gfap^+^*Sox2*^+^ (neural stem cells) populations and *Sox2*^+^ (progenitor) in newly performed experiments. As shown in revised Figure 4G and 4I, BrdU^+^Gfap^+^*Sox2*^+^ cells were similar in Tam-treated Nes Lrp4 CKO mice, compared with control, suggesting that Lrp4 knockout does not change the proliferation of neural stem cells at SC. However, BrdU^+^*Sox2*^+^ cells were reduced at SC by Lrp4 knockout, suggesting that Lrp4 is necessary for progenitor cell proliferation (revised Figure 4J). In either case, EE-induced increase in BrdU^+^ cells was attenuated by Lrp4 mutation. It suggests that at the basal level the effect of Lrp4 regulates adult neurogenesis more likely at the progenitor level.

3) The authors show in Figure 4C that there are significantly fewer labeled cells in iNestin-Lrp4f/f only 2 days after tamoxifen administration, but they do not give any explanation for this effect. This effect occurs in a strikingly small amount of time, so it suggests that there is either a proliferation defect in the rapidly amplifying progenitors or a cell death effect. The authors should address this issue at a minimum in the text, and potentially with additional experiments.

As shown in Figure 4—figure supplement 1F and 1G, cells positive for cleaved caspase-3 were similar between Nes Lrp4 CKO and control mice after two days of last Tam injection, suggesting Lrp4 knockout had no effect on apoptosis. We added the explanation for this effect may due to decrease the proliferation of progenitor, consistent with decrease the progenitor cell proliferation in Lrp4 CKOmice.